

# Vertical distribution of particle-phase dicarboxylic acids, oxoacids and α-dicarbonyls in the urban boundary layer based on the 325-meter tower in Beijing

**Wanyu Zhao[1,5,6], Hong Ren[2], Kimitaka Kawamura[3,5], Huiyun Du[1,6], Xueshun Chen[1,6], Siyao Yue[1,6], Qiaorong Xie[1,6], Lianfang Wei[1,6], Ping Li[1,6], Xin Zeng[4], Shaofei Kong[4], Yele Sun[1], Zifa Wang[1], and Pingqing Fu[2*]**

[1]State Key Laboratory of Atmospheric Boundary Layer Physics and Atmospheric Chemistry, Institute of Atmospheric Physics, Chinese Academy of Sciences, Beijing 100029, China

[2]Institute of Surface-Earth System Science, Tianjin University, Tianjin 300072, China

[3]Chubu Institute for Advanced Studies, Chubu University, Kasugai 487-8501, Japan

[4]Department of Atmospheric Science, School of Environmental Sciences, China University of Geosciences, Wuhan, 430074, China

[5]Institute of Low Temperature Science, Hokkaido University, Sapporo 060-0819, Japan

[6]College of Earth and Planetary Sciences, University of Chinese Academy of Sciences, Beijing 100049, China

*Correspondence to*: Pingqing Fu (fupingqing@tju.edu.cn)

**Abstract.** Vertical distribution of dicarboxylic acids, oxoacids, α-dicarbonyls and other organic tracer compounds in fine aerosols ($PM_{2.5}$) was investigated from the ground surface (8 m) to 260 m at a 325-meter meteorological tower in Beijing in the summer of 2015. Results showed that the concentrations of oxalic acid ($C_2$), the predominant diacid, were more abundant at 120 m ($210 \pm 154$ ng m$^{-3}$) and 260 m ($220 \pm 140$ ng m$^{-3}$) than those at the ground level ($160 \pm 90$ ng m$^{-3}$). Concentrations of phthalic acid (Ph) decreased with the increase of height, demonstrating that the vehicular exhausts at the ground surface was the main contributor. Positive correlations were noteworthy for $C_2$/total diacids with mass ratios of $C_2$ to main oxoacids (Pyr and $\omega C_2$) and α-dicarbonyls (Gly and MeGly) in polluted days ($0.42 \leq r^2 \leq 0.65$), especially at the ground level. In clean days, the ratios of carbon content in oxalic acid to water soluble organic carbon ($C_2$-C/WSOC) showed larger values at 120 m and 260 m than those at the ground surface. However, in polluted days, the $C_2$-C/WSOC ratio mainly reached its maximum at the ground level. These phenomena may indicate the enhanced contribution of aqueous-phase oxidation to oxalic acid in polluted days. Combined with the influence of wind field, total diacids, oxoacids and α-dicarbonyls decreased by 22% − 58% under the

control on anthropogenic activities during the 2015 Victory Parade period. Furthermore, the PMF results showed that the secondary formation routes (secondary sulfate formation and secondary nitrate formation) were the dominant contributors (37 − 44%) to organic acids, followed by biomass burning (25 − 30%) and motor vehicles (18 − 24%). In this study, the organic acids at the ground level were largely associated with local traffic emissions, while the long-range atmospheric transport

followed by photochemical aging contributed more to diacids and related compounds in the boundary layer over Beijing than the ground surface.

## 1 Introduction

Organic aerosols typically make up 20 − 50 % of the atmospheric fine aerosols (PM$_{2.5}$) mass at continental midlatitudes. And a large portion of organic aerosols is water-soluble, contributing 20 − 75% to carbon mass of aerosols emitted from

uncompleted combustion sources (Falkovich et al., 2005; Pathak et al., 2011; Graham et al., 2002). Low molecular weight (LMW) dicarboxylic acids and related compounds are abundant water-soluble organic compound classes in urban (Wang et al., 2012), mountainous (Kawamura et al., 2013; Cong et al., 2015), remote marine (Mochida et al., 2007; Wang et al., 2006c), and the Arctic (Kawamura et al., 2010; Kawamura et al., 1996) aerosols, and also in snow, rain and fog (Sempére and Kawamura, 1994; Kawamura et al., 2001).

Owing to high water-solubility and hygroscopicity, dicarboxylic acids play an important role in aerosol chemistry via atmospheric processing (e.g., iron catalyzed photolysis and secondary component formation) (Laskin et al., 2012; Drozd et al., 2014; Pavuluri and Kawamura, 2012) and in Earth's climate by enhancing hygroscopic behavior of aerosols to act as cloud condensation nuclei (Andreae and Rosenfeld, 2008; Bilde et al., 2015; Kanakidou et al., 2005). Dicarboxylic acids and related compounds are largely produced by secondary oxidation pathways, including photochemical (Kawamura and

Gagosian, 1987; Pavuluri et al., 2015) and aqueous oxidation (Carlton et al., 2007; Ervens and Volkamer, 2010) in the atmosphere. They can also be directly emitted from natural emissions, like marine plankton activities (Rinaldi et al., 2011; Tedetti et al., 2006), and anthropogenic sources including biomass burning (Legrand and Angelis, 1996; Narukawa et al., 1999), vehicular exhausts and fossil fuel combustion (Rogge et al., 1993; Kawamura and Kaplan, 1987).



Previous field measurements mostly focused on organic aerosols at the ground surface. Vertical measurements have been extensively conducted at a global scale since the early 2010s (Han et al., 2015; Hu et al., 2011; Andreae et al., 2012; Chi et al., 2013). The Amazon Tall Tower Observatory was established ~150 km northeast of the city of Manaus, Brazil for comprehensive studies of meteorology (Morton et al., 2014; Quesada et al., 2012; Sun et al., 2011), trace gases (Andreae et

al., 2012; Trebs et al., 2012), aerosol compositions (Andreae et al., 2012; Yáñez-Serrano et al., 2015; Yáñez-Serrano et al., 2018; Rizzo et al., 2013; Saturno et al., 2018), and ecology (Pöhlker et al., 2019; Quesada et al., 2012) to investigate long-term trends of the Amazonian hydrological and biogeochemical cycling linked with the human perturbation (Andreae et al., 2015). The 304 m tower of the Zotino Tall Tower Observatory in central Siberia serves as a basis for monitoring biogeochemical gases (Chi et al., 2013; Mikhailov et al., 2017; Winderlich et al., 2010), aerosol characteristics (Chi et al.,

2013; Heintzenberg et al., 2011; Mikhailov et al., 2017) and atmospheric transport (Mikhailov et al., 2017) at a wide range of spatial and temporal scales.

Beijing, the capital of China, is surrounded by highly industrialized and urbanized areas (Xia et al., 2007) and faces the severe haze pollution problems in the world (Wang et al., 2016). The haze events in China are characterized by regional distribution (Zhao et al., 2013) and are mainly driven by secondary aerosol formation, accounting for 30 – 77% of aerosol

mass and 44 – 71% of organic aerosol mass in fine mode (Huang et al., 2014). Field measurements in urban locations at the ground level are heavily influenced by local emissions (Huang et al., 2014), which brings uncertainties to quantify the relative contribution of regional transport to air quality. However, the vertical analysis can largely make up for the deficiency of surface observation and provide more physical and chemical information about the atmospheric process and structure. Guo et al. (2016) found that thermal stratification was clearly observed for urban boundary-layer in severe haze

events, and the vertical distributions of air pollutants were largely affected by the structure of atmospheric boundary layer. Guinot et al. (2006) reported that the Urban Canopy Layer (60 ~ 90 m) in Beijing is associated with the height of buildings and the mean width of street, both factors directly influencing local turbulence and pollutants dispersion. Furthermore, vertical measurements of extinction coefficient, gas precursors and aerosol compositions (e.g., organic compounds, sulfate

and nitrate) (Wang et al., 2018; Zhang et al., 2017) have also been conducted in detail at the 325 m meteorological tower in

Beijing to better understand the formation and evolution mechanisms of haze events. High loadings of dicarboxylic acids in

the atmosphere are closely linked to substantial anthropogenic activities (Kawamura et al., 2013; Wang et al., 2012; Wang et

al., 2006b). Understanding of the vertical characteristics of organic components in aerosols is important to elaborate the

oxidation mechanism, and their interactions with the lower boundary layer in haze events. The vertical investigation of

organic aerosols at a molecular level is far from complete, especially for diacids and related compounds.

This study investigates, for the first time, the vertical distributions of LMW dicarboxylic acids, oxoacids and α-dicarbonyls

in $PM_{2.5}$ collected at Beijing from 15[th] August to 10[th] September 2015, along with the analyses of ions, water-soluble organic

carbon (WSOC), organic carbon (OC), elemental carbon (EC) and tracer compounds like levoglucosan (biomass burning

tracer) and isoprene-oxidation products. The sources, aerosol chemistry, atmospheric long-range transport, and the effects of

emission control on anthropogenic activities were discussed.

## 2. Experimental methods

### 2.1 Aerosol sampling

The three-layer sampling was performed at the rooftop of a two-story building (the ground level, 8 m a.g.l.) and two

platforms (120 m and 260 m a.g.l.) on the 325-meter tower in the Institute of Atmospheric Physics, Chinese Academy of

Sciences (39°58'28''N, 116°22'16''E) (Fig. 1), a typical urban location influenced by traffic and cooking emissions in

Beijing (Sun et al., 2012). $PM_{2.5}$ samples were collected onto pre-heated (450°C, 6 hours) quartz-fiber filters (Pallflex) by

using three high-volume air samplers (TISCH, USA) at different heights. All of them were run at an airflow rate of 1.0

$m^3$/min for 23 hours during the 2015 Victory Parade period (from 15[th] August to 10[th] September, n = 77). The sampling time

was divided into the first non-restriction period, the restriction and the second non-restriction periods. Field blanks were

collected in each period at three sampling layers by placing filters on the samplers for half a minute without pumping. After

the campaign, all filters were stored at −20°C until analysis.

### 2.2 Analyses of diacids and related compounds

The quantitative determinations of diacids, oxoacids and α-dicarbonyls in $PM_{2.5}$ samples followed the analytical

measurements as described elsewhere (Kawamura and Ikushima, 1993; Kawamura and Bikkina, 2016). Briefly, 3.14 $cm^2$ of

each filter sample was extracted with Milli-Q water under ultrasonication for water-soluble organic acids. Next these extracts

were concentrated to dryness using a rotary evaporator under vacuum and reacted with 14% $BF_3$/n-butanol at 100°C for 1

hour. Finally, the derivatives were dissolved in n-hexane and analyzed them by using a gas chromatograph (GC, Agilent

6980) equipped with HP-5 column (0.2 mm × 25 m, 0.5 μm film thickness) and FID detector. The same analytical method

was also used for field blank filters. The concentrations of targeted organic acids in this study were corrected for the field

blanks, and their recoveries were > 85%.

### 2.3 Determinations of organic tracers

Small discs of filters were extracted with dichloromethane/methanol (2:1, v/v). Then the extracts were filtered through

quartz wool, concentrated under vacuum and blew to dryness. Derivatization was the reaction with N,O-bis-(trimethylsilyl)

trifluoroacetamide (BSTFA) containing pyridine (5:1) and 1% trimethylsilyl chloride at 70°C for 3 hours. Finally, the

derivatives were added with 30 μL of n-hexane and the internal standard of $C_{13}$ n-alkane (1.43 ng μ$L^{-1}$) before gas

chromatography/mass spectroscopy (GC/MS) measurement. GC/MS analysis was performed using a Hewlett-Packard model

7890A GC coupled to an Agilent model 5975C mass selective detector (MSD). The recoveries of levoglucosan and isoprene

SOA tracers were better than 80%, and both concentrations were corrected for the field blanks.

### 2.4 Ions, WSOC, OC and EC measurements

For the analysis of ions, a filter aliquot of each sample was extracted by ultrapure Milli-Q water under ultrasonication. Then

the extracts were determined for cations ($Na^+$, $K^+$, $NH_4^+$, $Ca^{2+}$, $Mg^{2+}$) and anions ($F^-$, $Cl^-$, $NO_2^-$, $SO_4^{2-}$, $NO_3^-$) by an Ion

Chromatography (Dionex Aquion, Thermo Scientific, America). Data of cations and anions were determined simultaneously.

The extraction step of WSOC is similar to that of ions, but the material of extraction bottle is glass. Next the extracts were

measured for WSOC by a Shimadzu TOC-V CPH total carbon analyzer with the limit of detection of 0.1 μgC m$^{-3}$ (Kawamura et al., 2013). OC and EC were determined using thermal optical reflectance (TOR) following the Interagency Monitoring of Protected Visual Environments (IMPROVE) protocol on a DRI Model 2001 Thermal/Optical Carbon Analyzer (Chow et al., 2005). The limit of detection for the carbon analysis is 0.8 and 0.4 μgC cm$^{-2}$ for OC and EC

respectively, with a precision better than 10% for total carbon (TC). Mass concentrations of ions, WSOC, OC and EC reported in the present study were all corrected for the field blanks.

**2.5 FLEXPART-WRF modeling**

The Lagrangian particle dispersion model FLEXPART-WRF can be used to quantify the impact of potential source regions (Brioude et al., 2013). Atmospheric particles were released at the height of 8 m at the sampling site every three hours a day.

Backward simulation was run for three days and the residence time of aerosols was calculated.

**2.6 Meteorological parameters**

The meteorological data, such as temperature (T), relative humidity (RH) and wind (wind speed and direction), were also obtained at same locations, except for the highest sampling site (280 m a.g.l.). The wind speed and direction at the ground level were almost same in field campaign (Fig. S1a). During the restriction period in Beijing, the organic compounds at 120

m were largely influenced by clean northern wind, while the aerosol particles at 280 m were mainly affected by the northwestern wind and accompanied by the influence of polluted southern and southeastern winds. Different from the second non-restriction period, the air quality at 120 m and 280 m were largely associated with southwestern wind from industrializations in the first non-restriction episode.

# 3. Results and Discussion

**3.1 OC and EC**

EC is a useful tracer for incomplete combustion emissions, including vehicle exhausts, biomass burning and fossil fuel combustion (Turpin and Huntzicker, 1991; Bond et al., 2013), while OC is either emitted from primary sources or produced

by secondary oxidation pathways in the atmosphere (Clarke et al., 2004). Previous studies found that relatively large OC/EC

ratio is associated with biomass burning (7.3), whereas lower value is linked to vehicular exhausts (1.1) (Sandradewi et al.,

2008). And Watson et al. (2001) reported the OC/EC ratio of 4.0 attributed to fossil fuel combustion.

Figure 2 shows the daily variations of OC, EC, OC/EC, SOC and POC, SOC/POC in $PM_{2.5}$ collected at the ground level, 120

m and 260 m in summer. During the field campaign, concentration level of EC was considerably lower than OC, but their

variation trends were similar (Fig. 2a & 2b). The average value of OC/EC ratios at ground surface and upper layers nearly

constant, which showed relative difference < 0.5 and a slightly higher mean value (7.5 ± 1.5) at 8 m (Table S1). Air quality

in the North China Plain is considerably influenced by biomass burning after summer harvest (Fu et al., 2012; Sun et al.,

2016; Desyaterik et al., 2013). The variations of OC/EC ratios in this study covered the value known for motor exhausts,

fossil fuel combustion and biomass burning sources (Fig. 2c). Good linear relationships were observed for OC with EC (0.59

$\leq r^2 \leq 0.78$) (Fig. 3), which indicated an important anthropogenic combustion source to OC.

Secondary organic carbon (SOC) is derived from various physical and chemical transformation processing, like gas/particle

partitioning of semi-volatile compounds (Hallquist et al., 2009). Owing to the complexities of SOC formation routes, there is

no valid direct analytical measurement to determine the atmospheric concentration of SOC. Primary organic carbon (POC) is

directly emitted from natural and anthropogenic emissions (Blando and Turpin, 2000). Because EC only originates from

primary emissions and is inert in the atmosphere, it is often used as a marker to estimate POC concentration in the

atmosphere (Turpin and Huntzicker, 1991). By this approach, the concentrations of SOC and POC can be evaluated from the

following equations (Turpin and Huntzicker, 1995; Strader et al., 1999):

$$POC = (OC/EC)_{min} \times EC \tag{1}$$

$$SOC = OC_{total} - POC \tag{2}$$

The $(OC/EC)_{min}$ ratios were the minimum OC/EC ratios calculated at each sampling height, and $OC_{total}$ were the mass

concentrations of OC. The concentration of POC calculated using the first equation means the primary carbonaceous

aerosols from fossil fuel combustion (Turpin and Huntzicker, 1995). In this study, the $(OC/EC)_{min}$ ratios were 5.0, 2.6 and

2.5 at the ground level, 120 m and 260 m, respectively. POC showed the largest mass concentration at the ground level ($4.7 \pm 2.3$ µg m$^{-3}$), while SOC were more abundant at 120 m ($5.1 \pm 2.9$ µg m$^{-3}$) and 260 m ($4.7 \pm 2.2$ µg m$^{-3}$) (Table 1). The SOC/POC ratios at 120 m ($1.8 \pm 0.79$) and 260 m ($1.9 \pm 0.92$) were higher than those at the ground level ($0.51 \pm 0.3$) (Table. S1), demonstrating that more photochemically aged aerosols accumulated at upper layers (Fig. 2f).

**3.2 Organic molecular characterization**

Table 2 shows the concentration ranges of dicarboxylic acids, oxoacids and α-dicarbonyls with the mean values. Mean concentrations of total diacids at 120 m ($370 \pm 255$ ng m$^{-3}$) and 260 m ($380 \pm 216$ ng m$^{-3}$) were almost the same, which were larger than that at the ground level ($285 \pm 143$ ng m$^{-3}$). Meanwhile, the relative abundances of total diacids in carbonaceous fractions (organic carbon and total carbon) reached maximum values at 260 m (Table S1). These vertical phenomena were

also observed for total oxoacids and α-dicarbonyls, suggesting that the photochemical aging of diacids and related compounds slightly increased at 260 m of the atmosphere.

The molecular distributions of dicarboxylic acids were characterized by the dominance of oxalic acid ($C_2$), followed by malonic ($C_3$) and succinic ($C_4$) acids (Fig. 4). Their concentrations were higher at upper layers with similar diurnal trends (Figs. S3a – c), illustrating $C_2$, $C_3$ and $C_4$ diacids derived from common primary sources and/or secondary oxidation

pathways. $C_3$ diacid is mainly formed via hydrogen abstraction of OH radicals on $C_4$ diacid, followed by the decarboxylation reaction (Kawamura and Ikushima, 1993). The mass concentration ratio of malonic acid to succinic acid is a useful marker to estimate the relative contribution of primary production and photochemical formation to organic aerosols. The $C_3/C_4$ ratio shows a characteristically lower value for primary emissions (e.g. vehicular exhausts: $0.25 - 0.44$) (Kawamura and Ikushima, 1993), whereas it has a larger value more than or equal to unity in aged aerosol (Kawamura and Sakaguchi, 1999). In this

paper, the mean $C_3/C_4$ ratio values at 260 m ($1.2 \pm 0.21$) were slightly larger than those at ground surface ($1.1 \pm 0.2$) and 120 m ($1.1 \pm 0.14$) (Table S1), implying that organic aerosols at 260 m were influenced by photochemical formation via regional transport.


Adipic acid ($C_6$), the major oxidation product of 2-methylglutaric ($iC_6$) from anthropogenic cyclic olefins (Hamilton et al., 2006; Müller et al., 2007) and intermediate of relatively long-chain diacids, was more abundant at upper heights, especially at 260 m (Fig. 4). Good correlations were observed for $C_6/iC_6$ with $C_6$/total diacids at the ground level ($r_1^2 = 0.66$) and 260 m ($r_3^2 = 0.94$) (Fig. S4a), which indicate that the photooxidation of cyclic olefins from anthropogenic sources is an important

contributor to adipic acid in Beijing. Azelaic acid ($C_9$) is a major oxidation product of unsaturated fatty acids (Matsunaga et al., 1999) from meal cooking (Rogge et al., 1991) (Stephanou and Stratigakis, 1993), biomass burning (Ballentine et al., 1998), marine plankton activities (Mochida et al., 2007) and terrestrial higher plants emissions (Ballentine et al., 1998). Meanwhile, Kawamura and Kaplan (1987) reported that vehicular emission is also an important source to azelaic acid. The relative abundance of $C_9$ in total diacids ($C_9$/total diacids) showed the largest value ($0.07 \pm 0.02$) at the ground level (Table

S1), implying that $C_9$ may be derived from the photooxidation of corresponding hydrocarbons from local vehicular emissions. Positive linear relationship was found between $C_6/C_9$ and $C_6$/total diacids at 120 m ($r_2^2 = 0.51$) and 260 m ($r_3^2 = 0.86$) (Fig. S4b), suggesting that the breakdown of $C_9$ carbon chain to form $C_6$ may enhance during its upward transport.

Phthalic acid (Ph) was the fourth abundant diacid in this study (Fig. 4), which can be emitted from motor vehicles and fossil fuel combustion or secondarily oxidized from aromatic hydrocarbons like naphthalene from coal and biofuel combustions

(Kautzman et al., 2010; Wang et al., 2006a). Mass concentrations of Ph decreased with the sampling heights, illustrating that vehicular exhausts at the ground level was a major source to phthalic acid. Terephthalic acid (tPh), an isomer of Ph, is a tracer of plastic burning in municipal wastes (Kawamura and Pavuluri, 2010; Simoneit et al., 2005). Contrary to the vertical distribution of Ph, the average concentration of tPh at the ground level ($12 \pm 10$ ng m$^{-3}$) was slightly lower than those at 120 m ($15 \pm 15$ ng m$^{-3}$) and 260 m ($13 \pm 11$ ng m$^{-3}$).

Oxocarboxylic acids, the intermediates of the oxidation of mono-carboxylic acids, can further be photochemically oxidized to form diacids (Warneck, 2003; Carlton et al., 2007). The glyoxylic acid ($\omega C_2$) is the dominant oxoacid, followed by pyruvic acid (Pyr) (Fig. 4). Both acids were more abundant at upper layers. Concentrations of total dicarbonyls varied from 1.8 ng m$^{-3}$ to 33 ng m$^{-3}$ with a maximum value ($10 \pm 7.4$ ng m$^{-3}$) at 260 m. Glyoxal (Gly) and methyglyoxal (MeGly), the

two smallest α-dicarbonyls, are mainly produced by the photooxidation of biogenic (Zimmermann and Poppe, 1996; Fick et al., 2004; Ervens et al., 2004) (e.g., isoprene and monoterpenes) and anthropogenic (Volkamer et al., 2001) (e.g., aromatics, acetone and acetylene) volatile organic compounds (VOCs). And both are important precursors to form less-volatile organic acids such as $\omega C_2$, Pyr (Lim et al., 2005). Concentrations of Gly and MeGly increased with the sampling heights, and their

concentration variations were similar to those of $\omega C_2$ and Pyr (Figs. S3d − g), illustrating that $\omega C_2$, Pyr and α-dicarbonyls had similar sources and/or formation pathways.

### 3.3 Restriction versus non-restriction periods

Owing to the regional scale of haze events in China, the Chinese government took measures to cut down anthropogenic emissions in Beijing and surrounding areas to ensure good air quality during the 2015 Victory Parade period. These strict

restrictions included the stopping construction and demolition activities, banning vehicles with odd and even plate numbers on alternate days, forbidding open burnings and shutting down factories and power plants in Tianjin City, Inner Mongolia Autonomous Region, Hebei, Shandong, Shanxi and Henan provinces. The restriction time (R) started from 20[th] August to 3[rd] September 2015. Before and after this period were defined as the first (N1) and second (N2) non-restriction periods, respectively. The concentration ratios of selected organic compounds during the restriction to non-restriction periods (R/N)

were calculated (Fig. S5). The ratio less than unity indicates that the control on the air quality effectively improves in Beijing. Both the primary incomplete combustion sources and the wind are two key influential factors (Liang et al., 2017; Xu et al., 2017). In this study, the R/N ratios of OC, EC, POC, total diacids, total oxoacids and dicarbonyls were lower than unity, but their R/N2 ratios were larger than corresponding R/N1 values (Fig. S5). These results showed that the improvement of air quality in Beijing was mainly influenced by the control on anthropogenic emissions, followed by the wind (Liang et al.,

2017; Xu et al., 2017). The lower concentrations of organic compounds in the second non-restriction period than those in the first non-restriction period were attributed to the more influence of clean northwesterly winds (Fig. S1a). The R/N ratios for OC/EC and SOC/POC were larger than or equal to unit due to the reduction of primary pollutants from incomplete burning activities under control measurements. Moreover, both R/N1 and R/N2 ratios for SOC/POC decreased with the sampling

heights, demonstrating that vehicular emissions at the ground level was an important factor to dicarboxylic acids. The reduction of vehicular emissions resulted in the relatively low concentration of POC and the highest SOC/POC ratio at the ground level in restriction period. The contribution of vehicular emissions to upper sampling layers decreased during the atmospheric upward transport.

Footprint regions of atmospheric particles are shown in Fig. 5. During the non-restriction periods, Beijing was dominated by regional transport from the south and southwest industrial areas. However, the footprint area of organic aerosols in restriction period was mainly located in the northeastern direction of Beijing, where was relatively clean. Combined with the wind field, the wind speeds and directions at the ground level showed little difference in the whole sampling time (Fig. S1a). The wind direction at 120 m was similar to that at 260 m in each non-restriction period, but wind speeds at 260 m were

larger than those at 120 m. In comparison to the second non-restriction period, the organic aerosols at 120 m and 260 m were mainly affected by southwesterly wind in the first non-restriction episode. As for the restriction period in Beijing, more clean air masses from the northern areas arrived at 120 m, while organic aerosols at 260 m were largely influenced by the northwesterly wind and accompanied by the influence of polluted southerly and southeasterly winds. Photochemical production of diacids and related compounds can occur in the atmospheric long-range transport.

Similarly, the R/N ratios for most diacids and related compounds were lower than unity (Fig. 6), especially for $\omega C_2$, Pyr, tPh and α-dicarbonyls. Owing to the influence of wind, the decreased level of main diacids, oxoacids and α-dicabonyls (20% − 69%) were stronger during the first non-restriction period than that (10% − 55%) in the second non-restriction period. These phenomena indicated that anthropogenic emissions largely contributed to diacids and related organic precursors in Beijing. Furthermore, the decreased orders of most diacids and related compounds were the ground level < 120 m < 260 m in

restriction period compared to the non-restriction periods (Table S2). This also supported the conclusion that a upward transport of vehicular emissions was existed, and organic aerosols at upper layers were more attributed to regional transport.


### 3.4 Possible formation pathways of organic acids

To better estimate the relative contribution of primary sources and photochemical transformation to diacids and related compounds, linear regression analyses for selected marker compounds (Fig. S2) and diagnostic ratios (Fig. 7) were employed in this study. Levoglucosan is an important tracer of biomass burning (Simoneit, 2002). The isoprene SOA tracers are defined as the sum of six oxidation products of isoprene, including 2-methylglyceric acid, $C_5$-alkene triols (*cis*-2-methyl-1,3,4-trihydroxy-1-butane, *trans*-2-methyl-1,3,4-trihydroxy-1-butene and 3-methyl-2,3,4-trihydroxy-1-butane), 2-methylthreitol and 2-methylerythritol (Claeys et al., 2004). Isoprene, the major biogenic volatile organic compound, is abundantly derived from plants emissions (Sharkey et al., 2007). Compared to total α-dicarbonyls, better correlations were found between isoprene SOA tracers and total diacids ($0.35 \leq r^2 \leq 0.46$) and oxoacids ($0.32 \leq r^2 \leq 0.48$), indicating that higher plants emissions contribute to diacids and related compounds to a certain extent in summer in Beijing. Levoglucosan only correlated well with total diacids ($r_2^2 = 0.51$, $r_3^2 = 0.41$), oxoacids ($r_2^2 = 0.53$, $r_3^2 = 0.43$) and α-dicarbonyls ($r_2^2 = 0.63$, $r_3^2 = 0.52$) at upper heights, demonstrating that biomass burning was a key source to organic aerosols.

The concentration ratio of relative abundance of $C_2$ in total diacids ($C_2$/total diacids) is known as a useful marker to assess the photochemical processing level, because $C_2$ is the end product mostly formed via the oxidation of longer carbon-chain diacids and other precursors in the atmosphere (Kawamura and Bikkina, 2016). The $C_2$/total diacids ratio enhanced with the increase of $C_2$-C/TC ($0.75 \leq r^2 \leq 0.8$), $C_3/C_4$ ($0.58 \leq r^2 \leq 0.8$) and $C_2/C_4$ ($0.26 \leq r^2 \leq 0.62$) ratios (Figs. 7a, c − d), suggesting a possible formation of oxalic acid from higher carbon number homologues and related compounds. However, there was no relationship between $(C_3–C_{12})$-C/TC and $C_2$/total diacids, which imply that the supply of longer-chain diacids may be faster than their degradation rates to produce oxalic acid in Beijing. Intermediate diacids can still be abundantly produced by oxidation of organic precursors during the atmospheric long-range transport. Meanwhile, $C_4$/total diacids ratio exhibited positive correlations with the $C_4/C_5$ ($0.28 \leq r^2 \leq 0.41$) and $C_4/C_6$ ($0.24 \leq r^2 \leq 0.48$) ratios (Figs. 7e − f), illustrating that glutaric and adipic acids may possibly photodegrade to form succinic acid. These results suggested that the photodegradation

of longer chain diacids contributed to the formation of lower diacids homologues after primary emissions, such as biomass

burning and vehicular emissions in Beijing.

The 325-m meteorological tower in Beijing is well equipped for studying the vertical structure of urban boundary layer

(UBL) and the vertical mechanism of organic compounds. Guo et al. (2016) found that the urban boundary layer often has a

significant thermal stratification in heavy haze periods, which shows the convective instability in daytime and the extreme

convective stability in nighttime. Meanwhile, the geometric parameters of wind speed vector and the efficiency of turbulent

transport also show more obvious diurnal variations. Concentrations of organic compounds are significantly affected by the

combined effect of source intensity, meteorological condition, and vertical structure of urban boundary layer.

Based on sampling records, $16 - 17^{th}$ and $29 - 30^{th}$ August and $7 - 8^{th}$ September were labeled as polluted episodes.

Previous studies reported that the photochemical oxidation of biogenic and anthropogenic VOCs results in semi-volatile

gaseous Gly and MeGly, which can partition into the aerosol phase enriched with liquid water content or cloud/fog droplets

(Volkamer et al., 2001; Zimmermann and Poppe, 1996; Fick et al., 2004; Ervens et al., 2004). In these transformations, $C_2$ is

an end product formed via photochemical oxidation of the key intermediates such as $\omega C_2$ and Pyr (Lim et al., 2005). Thus,

the ratios of $C_2/\omega C_2$, $C_2/Pyr$, $C_2/Gly$ and $C_2/MeGly$ were applied to better understand the aqueous oxidation mechanism of

organic matters.

Compared to clean days, the relatively strong aqueous-phase oxidation of related precursors ($\omega C_2$, Pyr, Gly and MeGly)

contributed to the accumulation of $C_2$ in polluted days. Positive correlations were noteworthy for $C_2/total$ diacids with

$C_2/Gly$ ($0.42 \leq r^2 \leq 0.58$) and $C_2/MeGly$ ($0.53 \leq r^2 \leq 0.65$) at three sampling heights, while good linear relationships for

$C_2/total$ diacids with $C_2/\omega C_2$ ($r_1^2 = 0.58$) and $C_2/Pyr$ ($r_1^2 = 0.5$) only existed at the ground level in polluted episodes (Fig. 8).

In contrast, no significant connections were found between relative abundance of $C_2$ in total diacids and its mass ratios with

four precursors in clean days. Therefore, the increased aqueous-phase oxidation may be a major source of oxalic acid. It is

worth noting that OH radical-initiated aqueous oxidation may dominate the production of secondary organic aerosol in

polluted days. The aqueous formation in cloud or wet aerosol is also an important pathway to diacids and related compounds (Carlton et al., 2006; Carlton et al., 2007; Ervens and Volkamer, 2010; Tan et al., 2010).

Aged organic aerosols are usually characterized by the larger contribution of oxalic acid to WSOC ($C_2$-C/WSOC). For example, $C_2$-C/WSOC ratio was higher in the photochemically aged aerosols collected at Hong Kong (6.8%) (Ho et al., 2011)

and Mount Hua (6.3%) (Meng et al., 2014) compared with the ratio (0.17%) in Ulaanbaatar aerosols that are significantly affected by substantial anthropogenic emissions (Jung et al., 2010). Due to the high temperature and relative humidity, the photochemical reaction is active in Hong Kong (Ho et al., 2011). Mount Hua is the highest mountain in central China and is a typically isolated site to investigate the atmospheric long-range transport of organic compounds (Meng et al., 2014). In contrast, diacids and related compounds in winter were mainly associated with uncontrolled wastes plastic burning, coal

power plants and vehicular emissions in Ulaanbaatar (Jung et al., 2010). Generally, in clean days, the $C_2$-C/WSOC ratio showed relatively large values at upper heights in this study (Fig. 9a). Moreover, in the transition from clean to polluted days, the $C_2$-C/WSOC ratio values at the ground level, 120 m and 260 m slightly increased.

However, in the more polluted days, $C_2$-C/WSOC ratios at the ground level were obviously higher than those at 120 m and 260 m owing to the accumulation of pollutants and moisture in ground surface atmosphere (Guinot et al., 2006). In

comparison to the moderately polluted events of 17[th] August (P1) and 8[th] September (P3), $C_2$-C/WSOC ratio maximized at 120 m (3.2%) in lightly polluted day of 29[th] August (P2). According to the concentrations of OC and EC, the strongest polluted event occurred on 8[th] September during the field campaign in Beijing. The $C_2$-C/WSOC ratio at the ground level in P3 (5.3%) was higher than that in P1 (4.7%), which may increase with an enhancement of the pollution. Furthermore, $C_2$-C/WSOC ratio was larger in the ground level (5.3%) followed by 120 m (2.4%) and 260 m (2.2%) in P3, demonstrating

that $C_2$-C/WSOC ratio may decrease with an increase of sampling heights. These phenomena may indicate that the moderately polluted days were favorable for aqueous formation of $C_2$ in the lower troposphere, especially at the ground level. Similarly, the $C_2$-C/OC ratios at three sampling layers were higher in polluted days than clean days in general (Fig. 9b). We

observed largest values of $C_2$-C/OC at higher levels of 120 m and 260 m, which may be caused by more accumulation of

POC from local anthropogenic emissions at the ground level (Fig. 2).

Different from the vertical distribution of $C_2$-C/WSOC ratios, the largest value of Ph-C/WSOC was mostly observed at the

ground level (0.70%), followed by 120 m (0.53%) and 260 m (0.45%) (Fig. 9c). In comparison to the ratio value in P2, the

large differences between Ph-C/WSOC ratio at the ground level and upper layers ($\geq 0.6\%$) in P1 and P3 also supported the

stagnant meteorological condition in the moderately polluted days. But the value of Ph-C/OC ratio at the ground level in

polluted days were lower than those in clean days (Fig. 9d), which may be caused by the accumulation of organic precursors,

like naphthalene. Unlike gas pollutants, high loadings of fine aerosol interact strongly with meteorological variables in the

planetary boundary layer (PBL). Both aerosol scattering and absorption reduce the amount of solar radiation reaching the

ground and thus reduce the sensible heat fluxes, which suppresses the development of PBL and further aggravate the

pollution level (Li et al., 2017). Such positive feedback is especially strong in heavy pollution events (Li et al., 2017), hence

the photochemical formation of Ph at the ground level may be not as effective as clean days.

Hydrated Gly and MeGly formed via the photooxidations of biogenic and anthropogenic VOCs can subsequently produce

$\omega C_2$ and Pyr, and ultimately generate $C_2$ (Ervens et al., 2004; Lim et al., 2005). In P1, only $\omega C_2$-C/WSOC ratio at the ground

level remarkably increased (Fig. 9g). But all the ratios of Pyr-C/WSOC, $\omega C_2$-C/WSOC, Gly-C/WSOC and MeGly-C/WSOC

were obviously larger at the ground level than those at upper layers in P3 (Figs. 9e, i, k). These results suggested that

aqueous oxidation pathway was an important factor to the formation of $C_2$, Pyr, $\omega C_2$ and $\alpha$-dicarbonyls in the more polluted

days. The ratios of $C_2$-C/WSOC, Pyr-C/WSOC, $\omega C_2$-C/WSOC, Gly-C/WSOC and MeGly-C/WSOC at three sampling levels

in transformation periods were divided by those in the moderately polluted days (P/T) to evaluate the importance of aqueous

formation. The transformation periods were defined as the day before haze days. The P1/T1 ratios of $C_2$-C/WSOC,

Pyr-C/WSOC, $\omega C_2$-C/WSOC, Gly-C/WSOC and MeGly-C/WSOC were lower than corresponding P3/T3 ratios (Table. S3),

implying that during the strongest polluted event in this study, aqueous formation may contribute more to the concentrations

of $C_2$, Pyr, $\omega C_2$, Gly and MeGly. In addition, orders of P3/T3 ratio all values were: ground level > 120 m > 260 m. The

vertical P3/T3 ratios for ωC$_2$-C/WSOC (the ground level: 5.2, 120 m: 1.8, 260 m: 1.4), Gly-C/WSOC (the ground level: 5.7, 120 m: 1.8, 260 m: 1.6) and MeGly-C/WSOC (the ground level: 5.8, 120 m: 2.0, 260 m: 1.5) were higher than those of C$_2$-C/WSOC and Pyr-C/WSOC. These phenomena implied that the aqueous formation of C$_2$, Pyr, ωC$_2$ and α-dicarbonyls may decrease with the sampling heights in the most polluted events. And the increasing level of aqueous formation of C$_2$ and

related precursors may be associated with the pollution strength in Beijing.

### 3.5 Source apportionment of organic acids using PMF analysis

Based on the data of organic tracers and ions, the positive matrix factorization (PMF, USEPA) was employed to estimate the relative contributions of primary sources and secondary formation pathways to diacids and related compounds in this study. The abundance, naming abbreviations and indicative sources of the tracer compounds were summarized in Table 3. Details

of model stability of the six-factor solution was provided in Table S4. The PMF-resolved source profiles for the six factors were shown in Figs. 10a − f. Each factor was identified according to the dominant species. Secondary sulfate formation was identified by SO$_4^{2-}$ and isoprene SOA tracers, which indicated ozonolysis, OH radical-initiated oxidation and aqueous processing. Secondary nitrate formation was identified by the dominance of NO$_3^-$ and isoprene SOA tracers, mainly representing the OH radical-initiated oxidation. Owing to the existence of two double bonds, isoprene is highly reactive and

is readily oxidized in the atmosphere by OH, NO$_3$ and O$_3$. Higher loading of the isoprene SOA tracers was observed in the factor of secondary sulfate formation than secondary nitrate formation, which may indicate more overlapping of oxidation pathways.

Meanwhile, in comparison to NO$_3^-$ ($r^2 \leq 0.23$), better correlations were found between isoprene SOA tracers and SO$_4^{2-}$ ($0.44 \leq r^2 \leq 0.67$) (Fig. S6), being consistent with the above conclusion. Biomass burning was identified by the dominant species

of levoglucosan and EC. Contributions of vehicle exhausts were identified by the dominance of hopanes (*αβ*-hopane, *αβS&R*-homohopane and *αβS&R*-bishomohopane) and EC. Plants emissions were identified by the dominance of isoprene SOA tracers. Because the isoprene SOA tracers are not only viewed as a representative of SOA tracers (Magda et al., 2004;Surratt et al., 2010) but also a marker of biogenic sources (Guenther et al., 2006), like terrestrial higher plants



emissions (Sharkey et al., 2007). Coal combustion was identified by dominant species of the PAHs with their molecular

weights of 276 (indeno-[1,2,3-*cd*]pyrene and benzo [*ghi*]perylene) and hopanes.

The PMF-resolved factor contributions to total species, and total diacids, oxoacids and α-dicarbonyls were shown in Fig.

10g−j. The secondary source (secondary sulfate formation and secondary nitrate formation) was the dominant contributor

(44%) to total species, followed by biomass burning (27%) and motor vehicles (14%). Similar factor distribution was also

observed for total diacids, oxoacids and α-dicarbonyls, but the contribution of motor vehicles enhanced, especially in total

oxoacids. The plant emission is a small contributor (5 − 8%) to organic compounds. In this study, the contributed fraction of

anthropogenic emissions (48 − 49%) to diacids and related compounds were slightly larger than that of secondary formation

pathways (37 − 44%).

**4. Conclusions**

Current knowledge on vertical distributions of dicarboxylic acids and related compounds in fine aerosol collected at Beijing

based on observations is very limited. The air pollution is characterized by regional distribution in China. Compared to the

ground measurements, the vertical studies can provide special insights into the photochemical mechanisms and regional

transport of organic aerosols. In this study, different from the vertical distribution of Ph, the organic acids mainly showed

higher values at 260 m and 120 m than those at the ground surface. The diacids and related compounds were largely

influenced by vehicular emissions at the ground level, whereas the atmospheric long-range transport was an important

contributor to organic compounds at upper layers. Unlike clean days, the relative contribution of aqueous formation to

dicarboxylic acids enhanced in polluted days, especially at the ground level. Moreover, the increasing level of aqueous

formation of $C_2$ and related precursors may be associated with the pollution strength in Beijing. Combined with the influence

of wind, mass concentrations of total diacids, oxoacids and α-dicarbonyls were largely cut down (22 − 58%) under the

control on anthropogenic emissions. In this paper, PMF analysis showed that the contributed fraction of anthropogenic

emissions (48 − 49%) to diacids and related compounds were slightly larger than that of secondary formation pathways (37 −

44%).



*Data availability*. The dataset for this paper is available upon request from the corresponding author (fupingqing@tju.edu.cn).

*Competing interests*. The authors declare that they have no conflict of interest.

*Author contributions.* Pingqing Fu designed this research. $PM_{2.5}$ samples were collected by Hong Ren. Laboratory analyses were performed by Wanyu Zhao and Hong Ren. The manuscript was written by Wanyu Zhao and Pingqing Fu with consultation from all other authors.

**Acknowledgments and Data**

This work was supported the National Natural Science Foundation of China (Grant Nos. 41625014 and 41961130384). The authors declare no competing financial interests. Supporting description of analyzing methods, figures and tables are provided in the supporting information. The data used in this manuscript are listed in the tables, figures, and supplementary materials.

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



**Table 1.** Vertical concentrations ($\mu g\ m^{-3}$) of OC, EC, SOC, POC and TC in $PM_{2.5}$ aerosols collected at Beijing.

| Species (Abbr.) | The ground level (n = 27) | | 120 m (n = 25) | | 260 m (n = 25) | |
|---|---|---|---|---|---|---|
| | Range | Mean/SD | Range | Mean/SD | Range | Mean/SD |
| OC | 3.1–15 | 6.6/2.5 | 2.8–18 | 8.2/4.4 | 2.5–16 | 7.7/3.5 |
| EC | 0.6–2.6 | 0.9/0.5 | 0.4–3.2 | 1.3/0.7 | 0.3–3.3 | 1.3/0.8 |
| SOC | 0.0–5.0 | 2.0/1.1 | 0.0–12 | 5.1/2.9 | 0.0–10 | 4.7/2.2 |
| POC | 2.8–13 | 4.7/2.3 | 1.1–8.2 | 3.2/1.9 | 0.77–8.2 | 3.2/2.0 |
| TC | 3.6–17 | 7.6/3.0 | 3.6–20 | 9.4/5.1 | 2.9–21 | 9.0/4.1 |





**Table 2.** Vertical concentrations (ng m$^{-3}$) of dicarboxylic acids, oxoacids and α-dicarbonyls collected in Beijing from August 15$^{th}$ to September 10$^{th}$, 2015.

| Species (Abbr.) | The ground level (n = 27) | | 120 m (n = 25) | | 260 m (n = 25) | |
|---|---|---|---|---|---|---|
| | Range | Mean/SD | Range | Mean/SD | Range | Mean/SD |
| Dicarboxylic acids | | | | | | |
| Oxalic, C$_2$ | 46–432 | 160/90 | 52–570 | 210/154 | 60–650 | 220/140 |
| Malonic, C$_3$ | 7.1–49 | 22/10 | 8.6–82 | 32/25 | 11–75 | 34/17 |
| Succinic, C$_4$ | 7.3–46 | 21/10 | 8.7–87 | 30/21 | 9.2–76 | 31/18 |
| Glutaric, C$_5$ | 3.2–17 | 6.5/3.2 | 2.8–29 | 9.2/6.4 | 3.2–25 | 9.5/5.2 |
| Adipic, C$_6$ | 2.5–14 | 5.3/2.5 | 3.7–21 | 8.2/4.2 | 4.8–21 | 13/5.0 |
| Pimeric, C$_7$ | 0.7–4.7 | 1.6/1.1 | 0.2–7.5 | 2.5/2.1 | 0.4–9.4 | 2.5/2.0 |
| Suberic, C$_8$ | BDL–0.3 | BDL/0.1 | BDL–0.8 | 0.1/0.2 | BDL–1.2 | 0.2/0.4 |
| Azelaic, C$_9$ | 12–34 | 18/4.8 | 4.9–53 | 21/10 | 5.6–38 | 17/7.6 |
| Decanedioic, C$_{10}$ | 0.4–3.1 | 1.3/0.7 | 0.2–4.2 | 1.6/1.0 | BDL–3.7 | 1.5/0.8 |
| Undecanedioic, C$_{11}$ | BDL–2.9 | 1.2/0.6 | 0.4–6.7 | 1.5/1.4 | 0.4–5.3 | 1.4/1.0 |
| Dodecanedioc, C$_{12}$ | BDL–0.2 | BDL | BDL–0.7 | 0.1/0.2 | BDL–0.7 | 0.2/0.2 |
| Methylmalonic, iC$_4$ | 0.4–2.8 | 0.9/0.5 | 0.2–4.1 | 1.2/0.8 | 0.5–2.3 | 1.1/0.5 |
| Methylsuccinic, iC$_5$ | 0.5–3.8 | 1.6/0.8 | 0.6–7.9 | 2.2/1.8 | 0.7–6.3 | 2.2/1.4 |
| 2-methylglutaric, iC$_6$ | BDL–1.7 | 0.5/0.4 | 0.3–4.8 | 1.0/0.9 | 0.3–3.9 | 1.0/0.8 |
| Maleic, M | 0.7–2.6 | 1.3/0.5 | 0.5–6.0 | 1.8/1.2 | 0.9–4.3 | 2.0/0.9 |
| Fumaric, F | 0.2–3.0 | 1.4/0.7 | 0.2–6.3 | 1.5/1.5 | 0.5–5.8 | 1.6/1.3 |
| Methylmaleic, mM | 0.7–3.0 | 1.3/0.5 | 0.5–7.4 | 1.9/1.6 | 0.6–4.2 | 1.9/1.0 |
| Phthalic, Ph | 8.3–61 | 26/11 | 8.6–51 | 23/11 | 8.2–40 | 21/7.9 |
| Isophthalic, iPh | 0.3–1.3 | 0.6/0.3 | 0.2–3.0 | 1.2/1.0 | BDL–8.4 | 1.0/1.7 |
| Terephthalic, tPh | 1.8–49 | 12/10 | 3.4–64 | 15/15 | 2.8–49 | 13/11 |
| Malic, hC$_4$ | BDL−1.7 | 0.3/0.3 | 0.2–2.4 | 0.6/0.5 | 0.1–2.4 | 0.7/0.5 |
| Oxomalonic, kC$_3$ | 0.5–9.6 | 2.7/2.2 | 0.6–13 | 4.0/3.4 | 0.6–12 | 4.1/2.5 |
| 4-oxopimelic, kC$_7$ | 0.5–9.7 | 3.0/2.2 | 0.4–13 | 4.4/3.6 | 1.1–12 | 5.0/2.6 |
| Total diacids | 99–733 | 285/143 | 110–945 | 370/255 | 126–1001 | 380/216 |
| Oxocarboxylic acids | | | | | | |
| Pyruvic, Pyr | 0.6–17 | 5.6/3.8 | 0.8–30 | 10/9.1 | 0.7–30 | 8.5/6.8 |
| Glyoxylic, ωC$_2$ | 0.5–44 | 15/11 | 2.6–80 | 21/19 | 2.5–65 | 20/17 |
| 3-oxopropanoic, ωC$_3$ | 0.9–8.2 | 2.9/1.8 | 1.3–11 | 4.1/3.1 | 1.0–12 | 3.7/2.4 |
| 4-oxobutanoic, ωC$_4$ | 1.0–11 | 4.5/2.5 | 2.7–21 | 7.3/4.4 | 1.9–17 | 6.9/3.8 |
| 5-oxopentanoic, ωC$_5$ | 0.3–3.4 | 1.5/0.7 | 0.6–4.7 | 2.0/1.1 | 0.7–3.9 | 1.9/0.9 |
| 7-oxoheptanoic, ωC$_7$ | 1.1–5.4 | 3.2/1.2 | 1.3–8.6 | 4.2/2.2 | 1.8–8.0 | 4.0/1.5 |
| 8-oxooctanoic, ωC$_8$ | 0.7–8.7 | 4.0/1.9 | 0.8–15 | 5.0/3.3 | 0.1–15 | 5.0/3.4 |
| 9-oxononanoic, ωC$_9$ | 0.2–2.4 | 1.2/0.6 | 0.3–4.9 | 1.2/1.0 | BDL–4.8 | 1.4/1.2 |
| Total oxoacids | 7.3–95 | 38/22 | 1.8–170 | 56/41 | 11–150 | 52/34 |
| α-dicarbonyls | | | | | | |
| Glyoxal, Gly | 0.5–6.1 | 2.2/1.2 | 1.0–13 | 3.3/2.7 | 0.8–9.8 | 3.4/2.3 |
| Methylglyoxal, MeGly | 1.4–12 | 4.1/2.6 | 1.0–20 | 5.2/4.0 | 1.3–22 | 6.7/5.2 |
| Total dicarbonyls | 1.8–17 | 6.3/3.7 | 2.2–33 | 8.5/6.6 | 2.0–31 | 10/7.4 |

BDL: below detection limit, which is ca. 0.005 ng m$^{-3}$ for the target compounds.



**Table 3.** Abundance and naming of measured ions ($\mu g\ m^{-3}$) and organic tracers (ng $m^{-3}$) used in the PMF analysis.

| Tracers | Grouping | Sources | Mean/SD | | |
|---|---|---|---|---|---|
| | | | The ground level | 120 m | 260 m |
| PAHs276 | indeno[1,2,3-cd]pyrene, benzo[ghi]perylene | Combustion sources (mainly coal combustion) | 0.41/0.23 | 0.24/0.18 | 0.08/0.06 |
| Levoglucosan | | Biomass burning | 19/16 | 21/14 | 23/15 |
| Hopanes | $\alpha\beta$-hopane, $\alpha\beta S\&R$-homohopane, $\alpha\beta S\&R$-bishomohopane | Fossil fuel combustion (e.g. vehicle exhaust, coal combustion) | 1.5/0.5 | 2.0/1.0 | 1.0/0.6 |
| Isoprene SOA tracers | 2-methylglyceric acid, 2-methylthreitol, 2-methylerythritol, $C_5$-alkene triols | Isoprene derived SOA, plants emissions | 31/20 | 41/38 | 48/37 |
| $SO_4^{2-}$ | | Secondary sulfate formation | 36/34 | 43/46 | 51/46 |
| $NO_3^-$ | | Secondary nitrate formation | 20/19 | 29/30 | 36/43 |



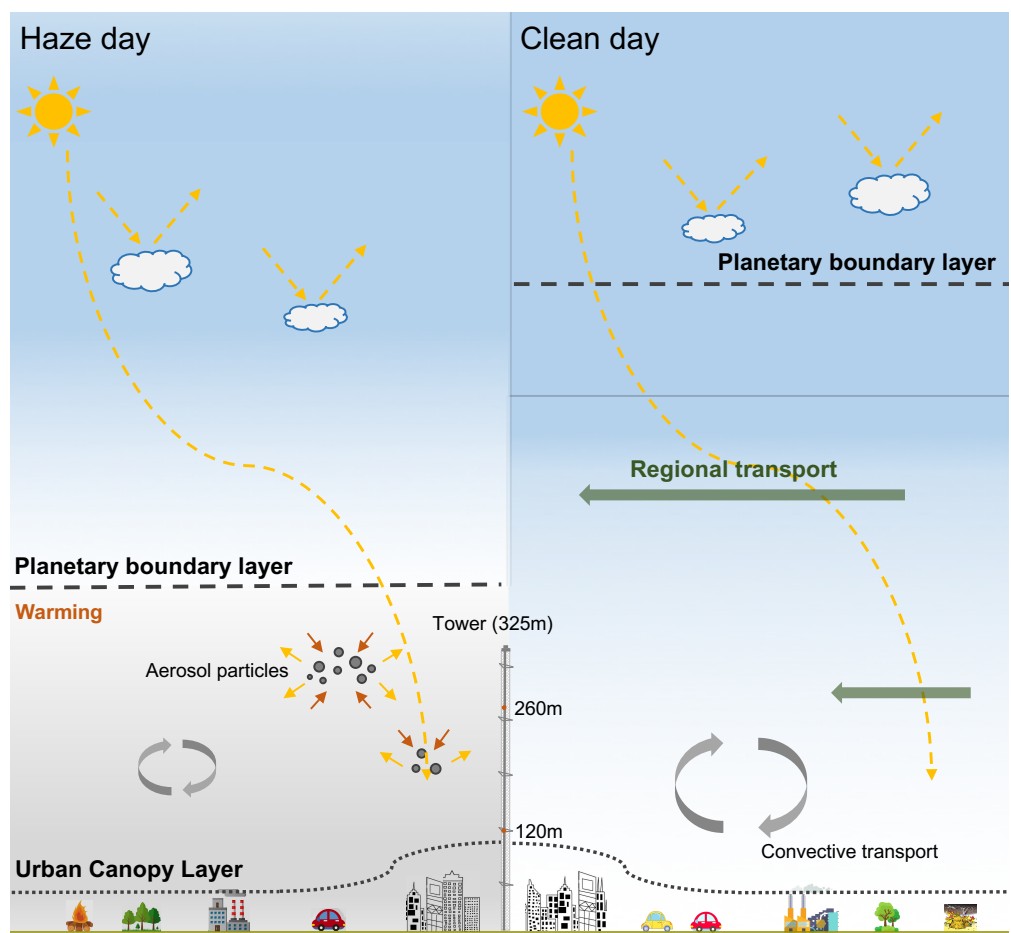

**Figure 1.** A schematic cartoon showing the atmospheric vertical structure over Beijing in polluted and clean days.

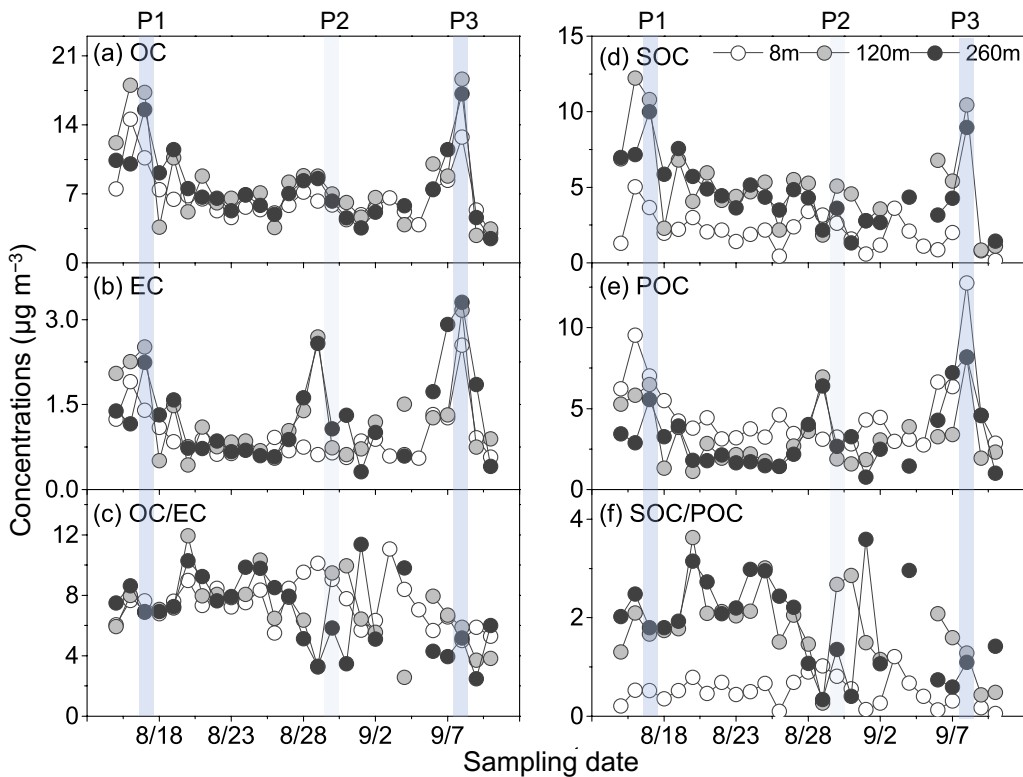

**Figure 2.** Daily variations in the concentrations of (a) OC, (b) EC, (c) OC/EC, (d) SOC, (e) POC and (f) SOC/POC at three sampling heights in Beijing. Two moderately polluted days (P1 and P3) and one lightly polluted day (P2) are marked for further discussions.





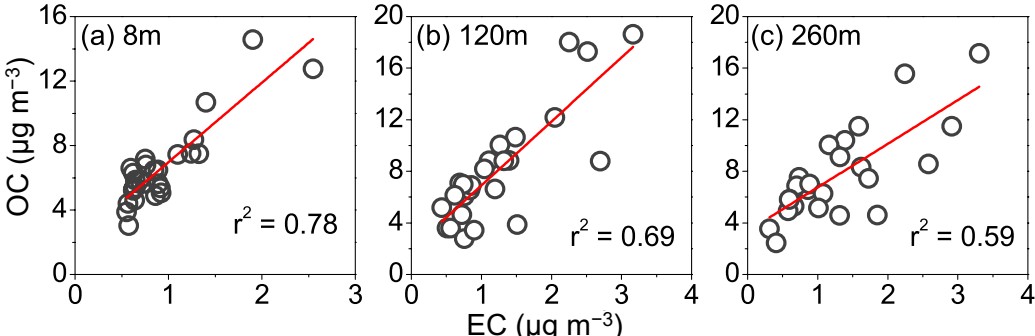

**Figure 3.** Correlations between OC and EC at the ground level, 120 m and 260 m.



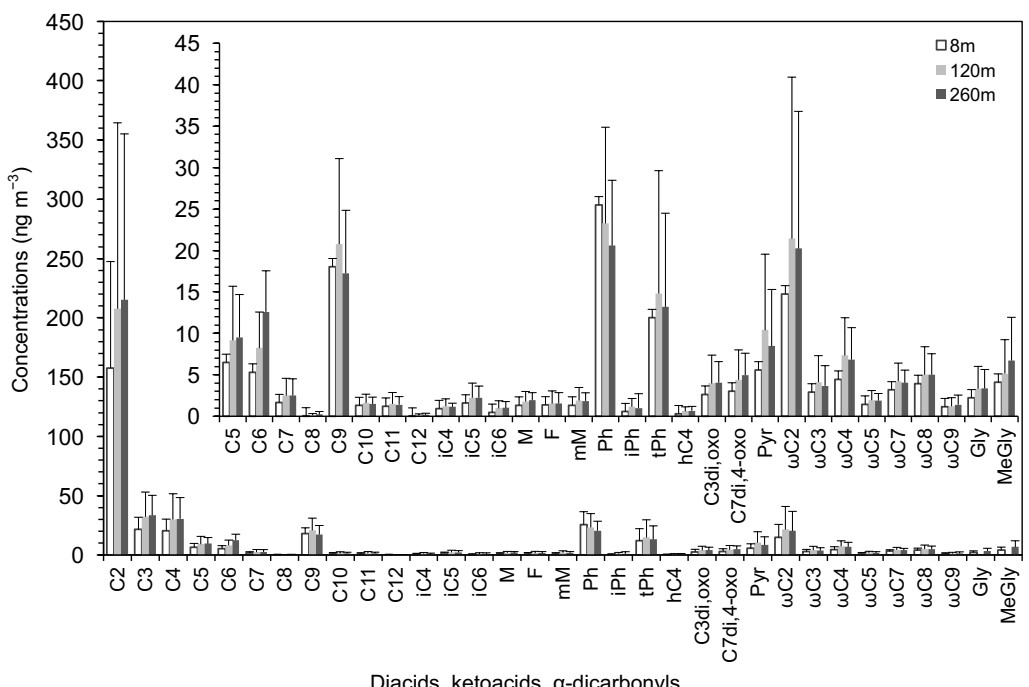

**Figure 4.** Molecular distributions of dicarboxylic acids and related compounds in the PM$_{2.5}$ samples collected at the tower from 15$^{th}$ August to 10$^{th}$ September 2015 in Beijing.



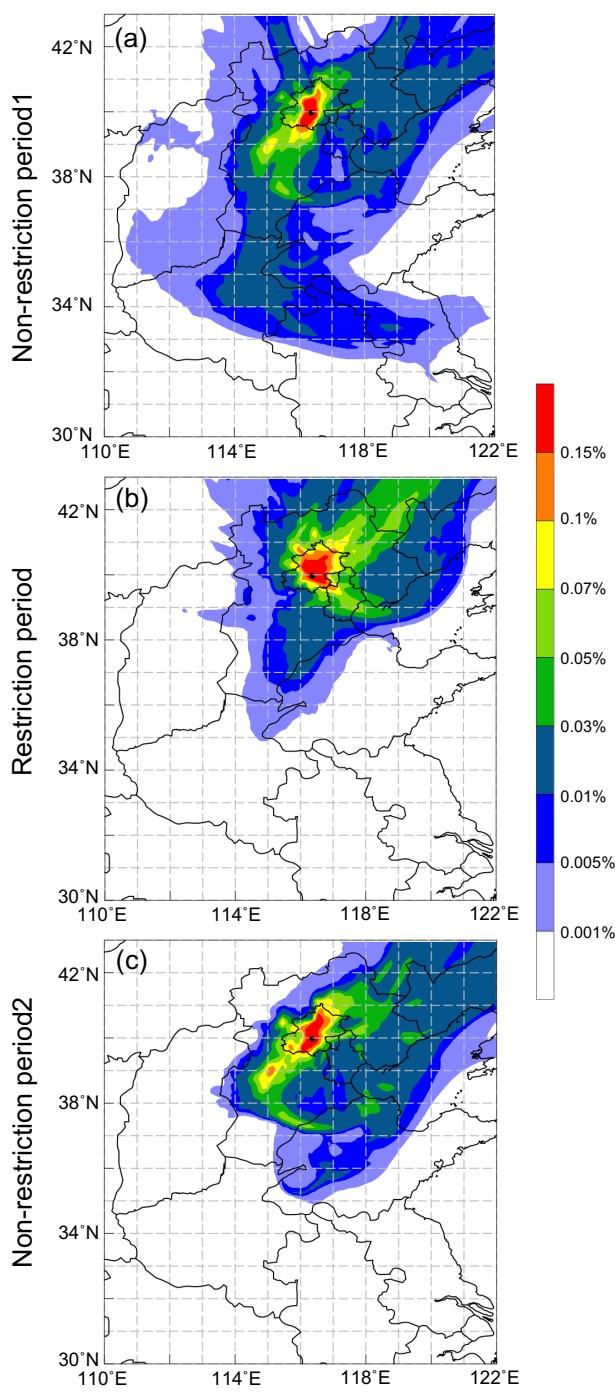

**Figure 5.** Aerosols footprint regions of non-restriction and restriction periods. The color bar indicates the relative residence time of tracer particles. And the black dot represents the sampling site.





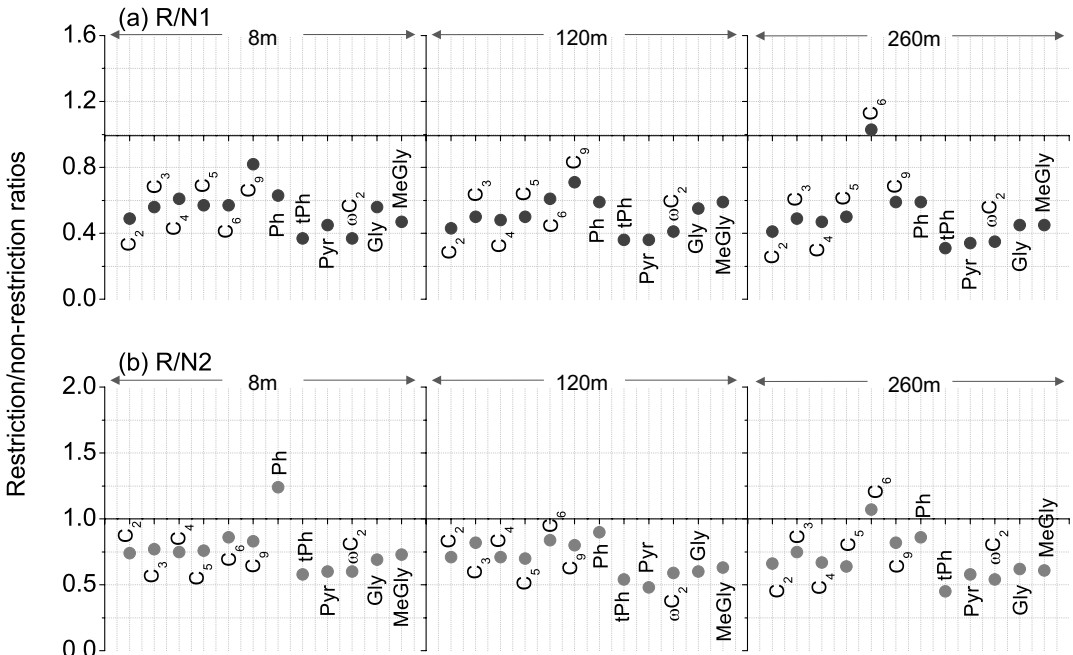

**Figure 6.** The R/N ratios of diacids and related compounds observed at the tower in Beijing in summer 2015.



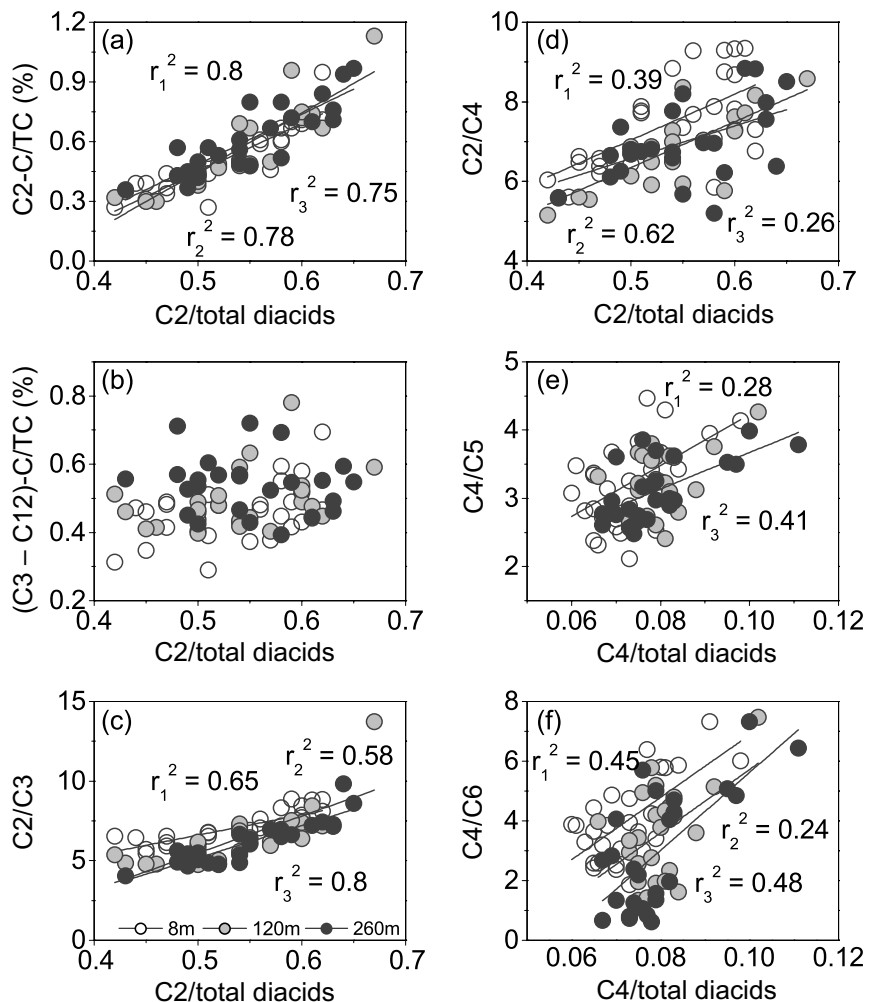

**Figure 7.** Contributions of (a) $C_2$-C/TC (%), (b) ($C_3$–$C_{12}$)-C/TC (%), (c) $C_2$/$C_3$ and (d) $C_2$/$C_4$ ratios as a function of relative abundance of $C_2$ in total diacids, as well as correlations for $C_4$/total diacids with (e) $C_4$/$C_5$ and (f) $C_4$/$C_6$. The $r_1^2$, $r_2^2$, and $r_3^2$ values are the correlation coefficients for those samples collected at 8 m, 120 m and 260 m, respectively.





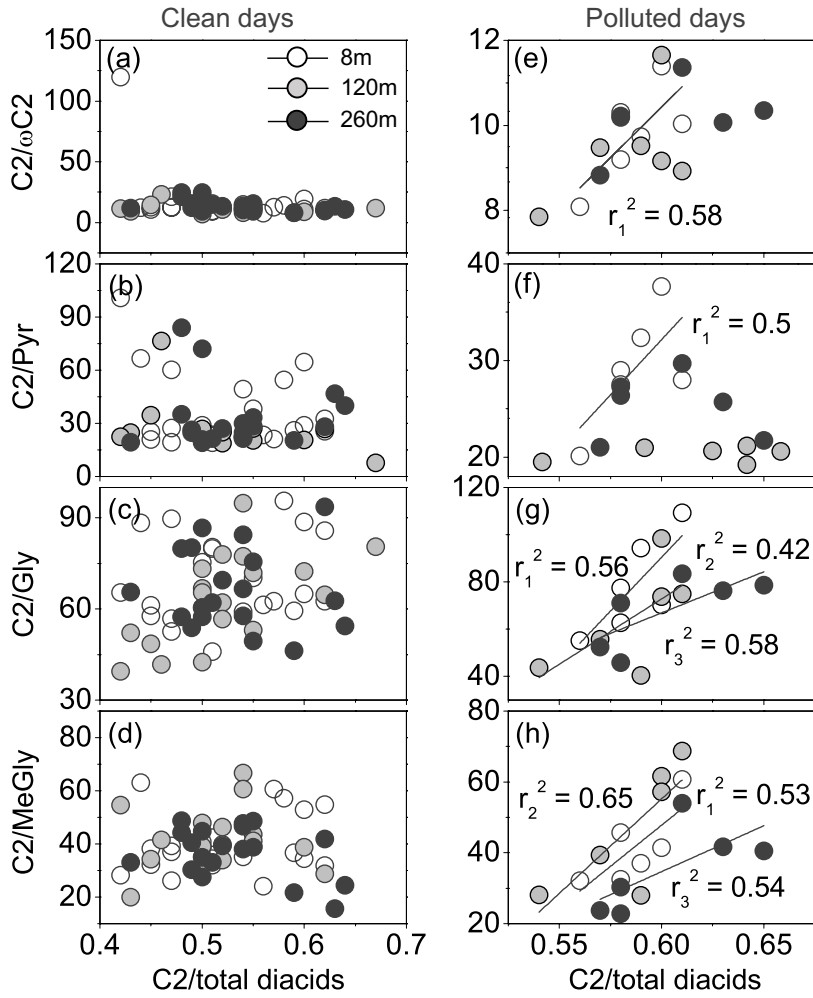

**Figure 8.** Correlations between concentration ratios of $C_2/\omega C_2$, $C_2/Pyr$, $C_2/Gly$, $C_2/MeGly$ and $C_2/total$ diacids in clean days and polluted episodes. The $r_1^2$, $r_2^2$, and $r_3^2$ values represent the correlation coefficients at 8 m, 120 m and 260 m, respectively.



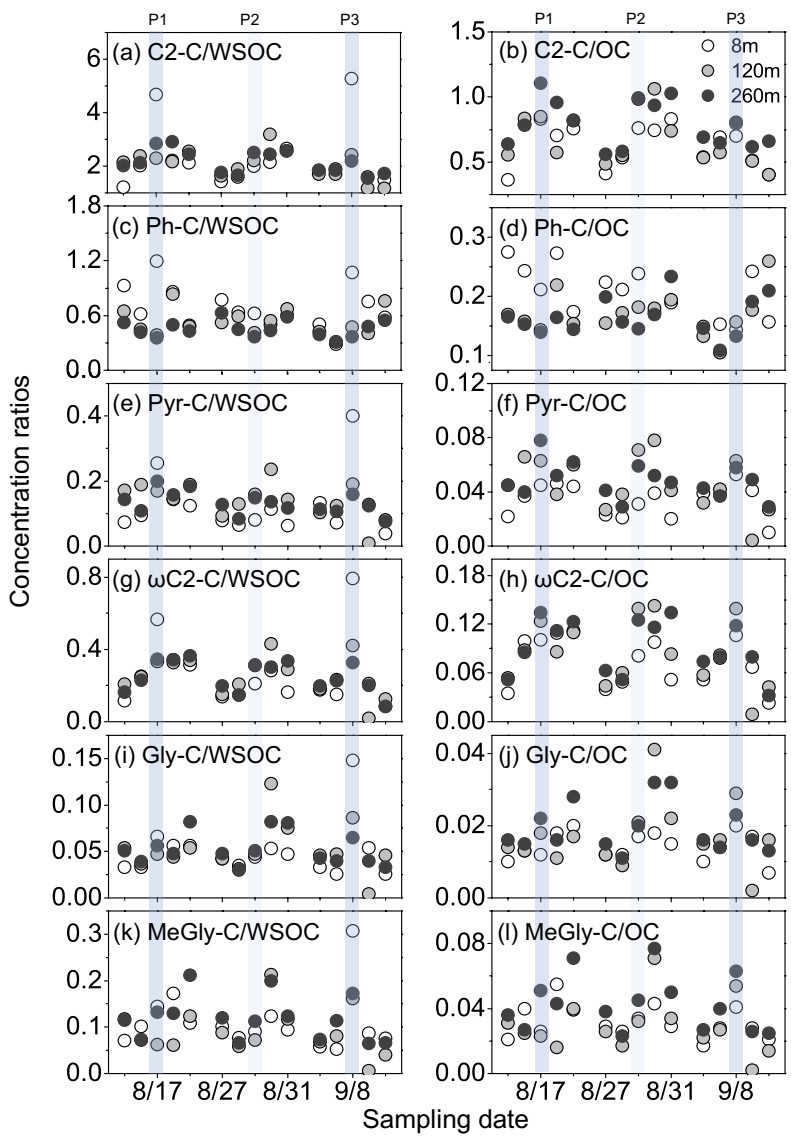

**Figure 9.** The relative contributions of C₂, Ph, ωC₂, Pyr, Gly and MeGly to carbonaceous fractions (WSOC and OC) in clean, transformation and polluted days. Two moderately polluted days (P1 and P3) and one lightly polluted day (P2) are marked for further discussions.



**Figure 10.** (a − f) Factor profiles (percentage of each species in factor) using ions and organics data for the six factors; (g − j) factor contributions to total species, diacids, oxoacids and α-dicarbonyls.