# Peer review of "Vertical distribution of particle-phase dicarboxylic acids, oxoacids and $\alpha$ -dicarbonyls in the urban boundary layer based on the 325-meter tower in Beijing"

_Atmospheric Chemistry and Physics, 2020_

## Referee Comment (RC1) · Anonymous Referee #1 · 9 Apr 2020

This paper investigates the vertical distribution of organic acids with altitude in Beijing based on measurements from a tower. The results show slightly higher concentrations of oxidized species aloft, not too surprising if the source are mainly from surface emissions which would mean less aged and less oxygenated. The data analysis is very detailed, but overall the results are only mildly interesting despite the measurements being very unique. This is because really no definite results are presented. The analysis is largely based on looking at ratios of species to infer processes or sources, and the interpretation of the results are always highly qualified with words such as, might,

and especially may. I wonder why there are not plots of altitude vs concentration in this paper, since that is fundamentally what the measurements were all about. It is not clear what the main contribution of the PMF source apportionment analyses is to the overall results of this paper on species vertical distributions; why was source apportionment not done at each altitude, was it because of insufficient data? The paper needs to be edited to improve the grammar. Overall, the data in this paper are interesting, but the analysis is very weak.

Pg 3 lines 12-13. Needs to be edited.

Please make clear that the high volume samplers were located at the elevation stated. That is, long sampling lines were not used in this study.

What type of filter was used, quartz, Teflon? Were the samples gas denuded (apparently not). There should be a discussion on possible artifacts associated with this sampling system, such as loss of the small acids due to evaporation, etc. Maybe this is why no light organic acids (eg, formate, acetate) are reported?

Page 7 lines 13 to 14. Stating that there is no direct analytical route to measure SOC may be strictly true, but there are ways that get pretty close, such as AMS measurements with PMF analysis. The statements and method to determine SOC in this paper are like reading a paper about 10 to 15 years old. Note the dates for the references for the method are 1995 and 1999. The reason this method is rarely used is that it is highly inaccurate. Limitations with the method need to be discussed.

---

## Referee Comment (RC2) · Anonymous Referee #2 · 13 Apr 2020

The manuscript studied the vertical distribution of particle-phase dicarboxylic acids, oxoacids, and α-dicarbonyls in urban Beijing during the 2015 Victory Parade period based on the 325-meter tower by using GC/MS, Ion Chromatography, and OC/EC analyzer. This study showed that concentrations of oxalic acid at 120 m and 260 m were more abundant than that at 8 m during the sampling period because of higher oxidation at high altitude. Vehicular exhausts were demonstrated as the main contributor for phthalic acid. Although this study did many correlation analyses in section 3, simple correlation analyses didn't effectively evidence the sources and formation of species

such as oxalic acid. In addition, many results and conclusions in section 3 were concluded with indefinite words such as may, causing that the conclusions were too speculative. Therefore, the authors should provide some stronger evidences in this section. This study further indicated the sources of organic acids based on meteorological parameters and FLEXPART-WRF model analyses. However, the source analysis at each sampling altitude was not showed. Is there any difference in the source of organic acids at each altitude? Because the study conducted on the vertical measurement, the data is valuable compared to the ground base. I might ask the author to make significant improvement on this paper before they can be accepted in the ACP.

P13 L21: How to exclude the contribution of accumulation of local emissions? Other evidences of aqueous-phase oxidation also should be provided. P4 L21: why were the blank samples only collected for half a minute? I saw that PM2.5 samples were collected for 23 h. P5 L6: Grammatically something wrong. P8 L4: How to exclude other sources (e.g., long-range transport, aqueous or heterogeneous reactions) for SOC? P8 L10: other sources should be discussed. Please see the above comment. P12 L16: C3/C4 should be C2/C3? P13 L9: What is the definition criteria of polluted episodes? P14 L15-16: Definition of pollution level should be provided. P17 L8: Was the coal combustion included in the anthropogenic emissions? Maybe need related literature? P17 L22: Same question as above.

---

## Author Comment (AC1) · 18 Jun 2020

**Responses to Reviewer #1**

We appreciate the reviewer for his/her thorough reading and thoughtful comments and suggestions, which greatly improve the quality of the manuscript. We revised the MS accordingly. The point-to-point responses to all the comments are given below in blue.

This paper investigates the vertical distribution of organic acids with altitude in Beijing based on measurements from a tower. The results show slightly higher concentrations of oxidized species aloft, not too surprising if the source are mainly from surface emissions which would mean less aged and less oxygenated. The data analysis is very detailed, but overall the results are only mildly interesting despite the measurements being very unique. This is because really no definite results are presented. The analysis is largely based on looking at ratios of species to infer processes or sources, and the interpretation of the results are always highly qualified with words such as, might, and especially may. I wonder why there are not plots of altitude vs concentration in this paper, since that is fundamentally what the measurements were all about. It is not clear what the main contribution of the PMF source apportionment analyses is to the overall results of this paper on species vertical distributions; why was source apportionment not done at each altitude, was it because of insufficient data? The paper needs to be edited to improve the grammar. Overall, the data in this paper are interesting, but the analysis is very weak.

**Response:** We thank the reviewer's careful reading and important comments, based on which we tried our best to improve the quality of our manuscript.

The ratios of organic compounds can be effectively applied to assessing the photochemical aging level and relative importance of primary emissions, because the ratio values vary with different sources and photochemical aging level of aerosols. Saxena and Hildemann (1996) and Seinfeld and Pandis (2006) reported that the fresh anthropogenic organic aerosols are mainly comprised by organic compounds with low oxidation level and hydrophobicity, such as alkanes, whereas aged particles and secondary organic aerosols have more hygroscopic compounds with oxygenated polar functional groups. The proportion of hydrophobic compounds in atmospheric particles decreases progressively when moving from the urban background to continental and to remote marine locations. Simultaneously, a population of more hygroscopic particles is always present (Kanakidou et al., 2005; Seinfeld and Pandis, 2006). For instance, the relative contents of total dicarboxylic acids in aerosol organic carbon (total diacids-C/OC) in remote

marine and polar areas ($\geq 10\%$) is higher than those in urban atmospheres ($1-3\%$) (Kawamura and Sakaguchi, 1999; Wang et al., 2006; Bikkina et al., 2015; Zhang et al., 2016).

Figure S3 has been redrawn for concentration variations and vertical properties of organic acids in the supporting information. Owing to the insufficient data, the PMF analysis run at each altitude existed high uncertainties, thus the PMF source apportionment analysis is to the overall results in this paper. The sample numbers will be considered to increase at each altitude. Simultaneously, more analyses will be employed to the aerosol samples to better estimate the relative contribution of sources and secondary process at different height.

This paper firstly shows the vertical distribution of organic compounds at the molecular level in megacity in China, investigates the sources and possible formation routes in clean and polluted days, and demonstrates the feedback of diacids and related compounds on the reduction of anthropogenic emissions. Such measurements in the troposphere are also critical for estimating the regional transport to air quality in Beijing and improving the simulation of aerosols in chemical transport models, which needs to be investigated in the future to give more precise conclusions. Moreover, the synchronous measurements, like Chemical Ionization Mass Spectrometer, is considered to use in combination with the sampling of diacids, oxoacids and α-dicarbonyls in gas and particle phase to better understand the gas-particle transformation of these acids at different heights.

Pg 3 lines 12-13. Needs to be edited.

**Response:** we have rephrased these sentences. Please see lines $12-15$ in page 3 in the revised manuscript.

Please make clear that the high volume samplers were located at the elevation stated. That is, long sampling lines were not used in this study.

**Response:** The samplers were located directly at the three layers, that is, the ground surface (8 m), 120 m, and 260 m at the 325-m meteorological tower. We have rephrased the sentences in the revised manuscript. Please see lines $18-19$ in page 4.

What type of filter was used, quartz, Teflon? Were the samples gas denuded (apparently not). There should be a discussion on possible artifacts associated with this sampling system, such as loss of the small acids due to evaporation, etc. Maybe this is why no light organic acids (eg, formate, acetate) are reported?

**Response:** We thank for the reviewer's suggestion. PM$_{2.5}$ samples were collected on quartz-fiber filters (Pallflex). Please see lines $17-18$ in page 4, where contains the filter information. Aerosol sampling onto quartz-fiber filters is accompanied by positive (e.g., adsorption of organic vapors) and/or negative (e.g., volatilization of organic aerosols after sample collection) artifacts. The collected particulate matter may chemically react with vapor components that pass through the filter during sampling to change the chemical composition of the deposit. This can result in positive or negative artifacts, especially if individual chemical compounds or chemical functionalities are being measured (Bennett and Stockburger, 1994). These sampling artifacts complicate the organic aerosol measurements, and it is difficult to separate the different competing artifact processes although this has been the object of many studies.

The positive artifact, as indicated by field blanks and backup filters, is believed to exceed the negative artifact for most samples (ten Brink et al., 2004; Watson et al., 2009). Without blank or backup filter subtraction, the artifact inflates organic carbon concentrations (Chow et al., 2010). Composition of the adsorbed/desorbed material, its exchange between gas and particle phases, the degree to which filters become saturated, and how the sign and amount of artifact differ among filter media and sampling environments have been studied, but these issues are not well understood (ten Brink et al., 2004; Watson et al., 2009; Vecchi et al., 2009; Arp et al., 2007; Zhu et al., 2012). In studies where the investigators attempted to measure the sampling error, the magnitudes of the determined positive or negative artifacts have varied considerably (Zhu et al., 2012; Liu et al., 2014a; Cheng et al., 2010a; Cheng et al., 2010b). This is probably due, in part, to differences in the organic molecular composition (hence in the relative volatilities and sorptivities) and in the concentrations that occurred at the different sampling locations. Multiple component sampling systems employing vapor phase separation devices such as denuders require a thorough evaluation to demonstrate efficient removal of the vapor at the operating conditions. This has been done for specific PAH compounds (Liu et al., 2014b; Schauer et al., 2003; Tsapakis and Stephanou, 2003; Liu et al., 2006). However, the application of their multiple component system to the determination of total OC aerosol for mixtures of atmospheric organic compounds presents a formidable challenge. The use of paired, parallel samplers (denuder difference method) is subject to the propagation of errors through subtraction, which degrades the precision. More development work with control experiments needs to be done on a wide range of organic compounds and concentrations before the sampling approach becomes credible for extended field studies.

Based on PM$_{10}$ aerosols collected using a high-volume sampler with two quartz fiber filters,

Ray and McDow (2005) found that dicarboxylic acids are affected by significant sampling artifact errors at concentrations below thresholds (0.5 ng m$^{-3}$ ≤ artifact thresholds ≤ 12 ng m$^{-3}$), but sampling artifacts are considerably lower at high concentrations expected when their primary emission sources and/or secondary formation are most important. Sihabut et al. (2005) also reported that the amount of dicarboxylic acids ($C_3 - C_9$) collected on the backup filter was consistently a small fraction of the amount collected on the front filter.

Due to the large contribution of primary emissions and secondary formation pathways to organic aerosols in Beijing, the concentrations of dicarboxylic acids detected in our study were larger than the artifact thresholds determined by Ray and McDow (2005). So far, most studies use single filter to collect organic and inorganic compounds in particulate aerosols (Wang et al., 2012; Zhang et al., 2015; Huang et al., 2014; Shi et al., 2018; Lyu et al., 2019; Mikhailov et al., 2017; Cheng et al., 2014; Ding et al., 2016; Cheng et al., 2016; Andreae et al., 2012; Saturno et al., 2018; Hsieh et al., 2008; Sanderson and Farant, 2005; Yue et al., 2019; Wang et al., 2016; Wang et al., 2017; An et al., 2019; Elser et al., 2016; Li et al., 2018; Kang et al., 2018; Liang et al., 2020; Zhang et al., 2016; Zhang et al., 2017; Wu et al., 2019; Zotter et al., 2014).

As for monocarboxylic acids, formic (HCOOH) and acetic (CH$_3$COOH) acids are among the most abundant and ubiquitous trace gases in the atmosphere (Khare et al., 1999; Talbot et al., 1988), and they have also been detected in remote, rural, polar, marine and urban aerosols, clouds and rain water (Chang et al., 2019; Andreae et al., 1988; Stavrakou et al., 2012; Paulot et al., 2011; Chattopadhyay et al., 2015; Chameides and Davis, 1983; Herndon et al., 2007; Zhang et al., 2011; Trentmann et al., 2005; Liu et al., 2012; Liang et al., 2020; Kesselmeier et al., 1998; Wang et al., 2007; Willey and Wilson, 1993; Avery et al., 2001; Xu et al., 2009; Kumar et al., 1996).

Our further work will focus on calculating the concentration thresholds of organic compounds in different locations and conditions and will try to find out more suitable sampling techniques to better understand the relationship between observed thresholds to adsorptive vapor concentrations and filter adsorption capacity for better evaluating the contribution of water-soluble organic acids to organic aerosol.

Page 7 lines 13 to 14. Stating that there is no direct analytical route to measure SOC may be strictly true, but there are ways that get pretty close, such as AMS measurements with PMF analysis. The statements and method to determine SOC in this paper are like reading a paper about 10 to 15 years old. Note the dates for the references for the method are 1995 and 1999.

The reason this method is rarely used is that it is highly inaccurate. Limitations with the method need to be discussed.

**Response:** Several indirect methods are widely employed to estimate the concentrations of primary and secondary organic aerosols, which can be categorized as the EC tracer method (Turpin and Huntzicker, 1995; Castro et al., 1999; Lim and Turpin, 2002; Chu, 2005; Yu et al., 2009), receptor modeling (Zheng et al., 2002; Na et al., 2004; Schauer et al., 2007) and chemical transport modeling (Pandis et al., 1992; Johnson et al., 2006b; Johnson et al., 2006a). Among these, the EC tracer method has received the widest application due to its simplicity and reliance on ambient measurements (Chu, 2005; Yu et al., 2009; Sun et al., 2011; Ji et al., 2016; Amato et al., 2011; Bougiatioti et al., 2013; Day et al., 2015; Grivas et al., 2012; Harrison and Yin, 2008; Kim et al., 2012; Kumar et al., 2012; Lin et al., 2009; Lonati et al., 2007; Samara et al., 2014; Saylor et al., 2006; Seguel A et al., 2009; Takegawa et al., 2006; Weber et al., 2007; Wu and Yu, 2016; Zhang et al., 2005; Zhang et al., 2014; Zhao et al., 2013; Zhou et al., 2012; Pio et al., 2011).

The EC tracer method is used here to derive POC and SOC empirically. The assumptions and methodology of EC tracer method are described in detail elsewhere (Castro et al., 1999; Turpin and Huntzicker, 1991, 1995; Yu et al., 2007). Briefly, total OC is defined as the sum of POC and SOC. By this method, the POC concentration can be defined as

$$POC = (OC/EC)_{pri} \times EC + c, \hspace{3cm} (1)$$

where $(OC/EC)_{pri}$ is the estimated primary OC/EC ratio and c is to account for non-combustion sources contributing to POC (Turpin and Huntzicker, 1995; Castro et al., 1999). The SOC concentration can be estimated as

$$SOC = OC - POC \hspace{4cm} (2)$$

Previous studies often summarized the linear least-squares fit results of OC vs. EC from the lowest 5% and 10% OC/EC values to estimate $(OC/EC)_{pri}$ (Sun et al., 2011; Miyazaki et al., 2006; Chu, 2005; Day et al., 2015; Ji et al., 2016), which is seasonally-dependent (Yuan et al., 2006). Therefore, the $(OC/EC)_{pri}$ should be calculated at each sampling height (8 m, 120 m and 260 m) in our study due to the different influence strength of primary sources and secondary formation pathways. But it is worth mentioning that the number of $PM_{2.5}$ samples collected at each sampling height were less than 30 in our study, which means that there is no statistical significance for the $(OC/EC)_{pri}$ estimation from the lowest 5% and 10% OC/EC values. The $(OC/EC)_{pri}$ is 0.21, 0.14 and 0.24 calculated from the lowest 20% OC/EC values at the ground surface, 120 m and 260 m, respectively (Fig. 1). Based on these $(OC/EC)_{pri}$ values, the possible

uncertainties (±138% − ±207%) largely vary for the estimated SOC. Meanwhile, these (OC/EC)$_{pri}$ ratios are not comparable to those (1.9 − 2.8) based on the lowest 5% OC/EC values in four seasons in Beijing, China (Ho et al., 2007; Ji et al., 2016).

The direct emission from vegetation, the major non-combustion source, contributed 3.8% to OC in total suspended particles in summer in Beijing (Li et al., 2018), so OC emitted from non-combustion sources is assumed to be negligible in the approach used here. Using the minimum OC to EC ratio, (OC/EC)$_{min}$, to substitute for (OC/EC)$_{pri}$, the SOC and POC can therefore be estimated (Yu et al., 2009; Cabada et al., 2004; Castro et al., 1999),. The (OC/EC)$_{min}$ ratios were 5.0, 2.6 and 2.5 at the ground level, 120 m and 260 m in summer in Beijing, respectively. The sampling site is a typical urban location largely influenced by traffic and cooking emissions in Beijing, especially at the ground surface (Zhou et al., 2018b; Zhou et al., 2018a). Additionally, cooking emission also correspond to high OC/EC ratios (Samara et al., 2014). Thus, the (OC/EC)$_{min}$ ratio at the ground surface (5.0) was larger than those at 120 m (2.6) and 260 m (2.5) in our study and those (1.9 − 2.8) detected in other sampling sites in Beijing (Ho et al., 2007; Ji et al., 2016). It is rational to use the minimum OC/EC ratio to estimate the SOC and POC concentrations in this paper.

[Figure]

Figure 1. The linear least-squares analysis fitting results grouped by the lowest 20% of OC/EC ratios at the ground level, 120 m and 260 m

Based on AMS online data, the PMF result does get closer to the true SOC value, but this method is not widely used and is beyond the scope of our study. Because our group focus on the offline analyses of aerosol chemistry at the molecular level.

The 1995 and 1999 references cited for the EC tracer method in our study are the most classic literatures, which have been widely cited. Please see the lines 18 − 19 in page 7, where we have updated the references.

**References**

Amato, F., Viana, M., Richard, A., Furger, M., Prévôt, A. S. H., Nava, S., Lucarelli, F., Bukowiecki, N., Alastuey, A., Reche, C., Moreno, T., Pandolfi, M., Pey, J., and Querol, X.: Size and time-resolved roadside enrichment of atmospheric particulate pollutants, Atmos. Chem. Phys., 11, 2917-2931, doi: 10.5194/acp-11-2917-2011, 2011.

An, Z., Huang, R.-J., Zhang, R., Tie, X., Li, G., Cao, J., Zhou, W., Shi, Z., Han, Y., Gu, Z., and Ji, Y.: Severe haze in northern China: a synergy of anthropogenic emissions and atmospheric processes, Proc. Nat. Acad. Sci. U.S.A., 116, 8657, doi: 10.1073/pnas.1900125116, 2019.

Andreae, M. O., Talbot, R. W., Andreae, T. W., and Harriss, R. C.: Formic and acetic acid over the central Amazon region, Brazil: 1. Dry season, J. Geophys. Res. Atmos., 93, 1616-1624, doi: 10.1029/JD093iD02p01616, 1988.

Andreae, M. O., Artaxo, P., Beck, V., Bela, M., Freitas, S., Gerbig, C., Longo, K., Munger, J. W., Wiedemann, K. T., and Wofsy, S. C.: Carbon monoxide and related trace gases and aerosols over the Amazon Basin during the wet and dry seasons, Atmos. Chem. Phys., 12, 6041-6065, doi: 10.5194/acp-12-6041-2012, 2012.

Arp, H. P. H., Schwarzenbach, R. P., and Goss, K.-U.: Equilibrium sorption of gaseous organic chemicals to fiber filters used for aerosol studies, Atmos. Environ., 41, 8241-8252, https://doi.org/10.1016/j.atmosenv.2007.06.026, 2007.

Avery, G. B., Tang, Y., Kieber, R. J., and Willey, J. D.: Impact of recent urbanization on formic and acetic acid concentrations in coastal North Carolina rainwater, Atmos. Environ., 35, 3353-3359, https://doi.org/10.1016/S1352-2310(00)00328-9, 2001.

Bennett, R. L., and Stockburger, L.: Sampling carbonaceous aerosols: A review of methods and previous measurements. Report for December 1993-May 1994, Environmental Protection Agency, Research 1994.

Bikkina, S., Kawamura, K., and Miyazaki, Y.: Latitudinal distributions of atmospheric dicarboxylic acids, oxocarboxylic acids, and α-dicarbonyls over the western North Pacific: Sources and formation pathways, J. Geophys. Res. Atmos., 120, 5010-5035, doi: 10.1002/2014jd022235, 2015.

Bougiatioti, A., Zarmpas, P., Koulouri, E., Antoniou, M., Theodosi, C., Kouvarakis, G., Saarikoski, S., Mäkelä, T., Hillamo, R., and Mihalopoulos, N.: Organic, elemental and water-soluble organic carbon in size segregated aerosols, in the marine boundary layer of the Eastern Mediterranean, Atmos. Environ., 64, 251-262, https://doi.org/10.1016/j.atmosenv.2012.09.071, 2013.

Cabada, J. C., Pandis, S. N., Subramanian, R., Robinson, A. L., Polidori, A., and Turpin, B.: Estimating the secondary organic aerosol contribution to PM2.5 using the EC Tracer method special issue of aerosol science and technology on findings from the fine particulate matter supersites program, Aerosol Sci. Technol., 38, 140-155, doi: 10.1080/02786820390229084, 2004.

Castro, L. M., Pio, C. A., Harrison, R. M., and Smith, D. J. T.: Carbonaceous aerosol in urban and rural European atmospheres: estimation of secondary organic carbon concentrations, Atmos. Environ., 33, 2771-2781, https://doi.org/10.1016/S1352-2310(98)00331-8, 1999.

Chameides, W. L., and Davis, D. D.: Aqueous-phase source of formic acid in clouds, Nature, 304, 427-429, doi: 10.1038/304427a0, 1983.

Chang, D., Wang, Z., Guo, J., Li, T., Liang, Y., Kang, L., Xia, M., Wang, Y., Yu, C., Yun, H., Yue, D., and Wang, T.: Characterization of organic aerosols and their precursors in southern China during a severe haze episode in January 2017, Sci. Tot. Environ., 691, 101-111, https://doi.org/10.1016/j.scitotenv.2019.07.123, 2019.

Chattopadhyay, A., Chatterjee, P., and Chakraborty, T.: Photo-oxidation of acetone to formic acid in synthetic air and its atmospheric implication, J. Phys. Chem. A, 119, 8146-8155, doi: 10.1021/acs.jpca.5b04905, 2015.

Cheng, Y., He, K. B., Duan, F. K., Zheng, M., Ma, Y. L., Tan, J. H., and Du, Z. Y.: Improved measurement of carbonaceous aerosol: evaluation of the sampling artifacts and inter-comparison of the thermal-optical analysis methods, Atmos. Chem. Phys., 10, 8533-8548, doi: 10.5194/acp-10-8533-2010, 2010a.

Cheng, Y., Lee, S. C., Ho, K. F., and Fung, K.: Positive sampling artifacts in particulate organic carbon measurements in roadside environment, Environ. Monit. Assess., 168, 645-656, 10.1007/s10661-009-1140-1, 2010b.

Cheng, Y., Engling, G., He, K.-b., Duan, F.-k., Du, Z.-y., Ma, Y.-l., Liang, L.-l., Lu, Z.-f., Liu, J.-m., and Zheng, M.: The characteristics of Beijing aerosol during two distinct episodes: Impacts of biomass burning and fireworks, Environ. Pollut., 185, 149-157, 2014.

Cheng, Y., Zheng, G., Wei, C., Mu, Q., Zheng, B., Wang, Z., Gao, M., Zhang, Q., He, K.,

Carmichael, G., Pöschl, U., and Su, H.: Reactive nitrogen chemistry in aerosol water as a source of sulfate during haze events in China, Sci. Adv., 2, e1601530, doi: 10.1126/sciadv.1601530, 2016.

Chow, J. C., Watson, J. G., Chen, L. W. A., Rice, J., and Frank, N. H.: Quantification of PM2.5 organic carbon sampling artifacts in US networks, Atmos. Chem. Phys., 10, 5223-5239, doi: 10.5194/acp-10-5223-2010, 2010.

Chu, S.-H.: Stable estimate of primary OC/EC ratios in the EC tracer method, Atmos. Environ., 39, 1383-1392, https://doi.org/10.1016/j.atmosenv.2004.11.038, 2005.

Day, M. C., Zhang, M., and Pandis, S. N.: Evaluation of the ability of the EC tracer method to estimate secondary organic carbon, Atmos. Environ., 112, 317-325, https://doi.org/10.1016/j.atmosenv.2015.04.044, 2015.

Ding, A. J., Huang, X., Nie, W., Sun, J. N., Kerminen, V. M., Petäjä, T., Su, H., Cheng, Y. F., Yang, X. Q., Wang, M. H., Chi, X. G., Wang, J. P., Virkkula, A., Guo, W. D., Yuan, J., Wang, S. Y., Zhang, R. J., Wu, Y. F., Song, Y., Zhu, T., Zilitinkevich, S., Kulmala, M., and Fu, C. B.: Enhanced haze pollution by black carbon in megacities in China, Geophys. Res. Lett., 43, 2873-2879, doi: 10.1002/2016GL067745, 2016.

Elser, M., Huang, R. J., Wolf, R., Slowik, J. G., Wang, Q., Canonaco, F., Li, G., Bozzetti, C., Daellenbach, K. R., Huang, Y., Zhang, R., Li, Z., Cao, J., Baltensperger, U., El-Haddad, I., and Prévôt, A. S. H.: New insights into PM2.5 chemical composition and sources in two major cities in China during extreme haze events using aerosol mass spectrometry, Atmos. Chem. Phys., 16, 3207-3225, doi: 10.5194/acp-16-3207-2016, 2016.

Grivas, G., Cheristanidis, S., and Chaloulakou, A.: Elemental and organic carbon in the urban environment of Athens. Seasonal and diurnal variations and estimates of secondary organic carbon, Sci. Tot. Environ., 414, 535-545, https://doi.org/10.1016/j.scitotenv.2011.10.058, 2012.

Harrison, R. M., and Yin, J.: Sources and processes affecting carbonaceous aerosol in central England, Atmos. Environ., 42, 1413-1423, https://doi.org/10.1016/j.atmosenv.2007.11.004, 2008.

Herndon, S. C., Zahniser, M. S., Nelson Jr, D. D., Shorter, J., McManus, J. B., Jiménez, R., Warneke, C., and de Gouw, J. A.: Airborne measurements of HCHO and HCOOH during the New England Air Quality Study 2004 using a pulsed quantum cascade laser spectrometer, J. Geophys. Res. Atmos., 112, doi: 10.1029/2006JD007600, 2007.

Ho, K. F., Cao, J. J., Lee, S. C., Kawamura, K., Zhang, R. J., Chow, J. C., and Watson, J. G.: Dicarboxylic acids, ketocarboxylic acids, and dicarbonyls in the urban atmosphere of China, J.

Geophys. Res., 112, 10.1029/2006jd008011, 2007.

Hsieh, L.-Y., Chen, C.-L., Wan, M.-W., Tsai, C.-H., and Tsai, Y. I.: Speciation and temporal characterization of dicarboxylic acids in PM2.5 during a PM episode and a period of non-episodic pollution, Atmos. Environ., 42, 6836-6850, 10.1016/j.atmosenv.2008.05.021, 2008.

Huang, R.-J., Zhang, Y., Bozzetti, C., Ho, K.-F., Cao, J.-J., Han, Y., Daellenbach, K. R., Slowik, J. G., Platt, S. M., and Canonaco, F.: High secondary aerosol contribution to particulate pollution during haze events in China, Nature, 514, 218-222, 2014.

Ji, D., Zhang, J., He, J., Wang, X., Pang, B., Liu, Z., Wang, L., and Wang, Y.: Characteristics of atmospheric organic and elemental carbon aerosols in urban Beijing, China, Atmos. Environ., 125, 293-306, 10.1016/j.atmosenv.2015.11.020, 2016.

Johnson, D., Utembe, S. R., and Jenkin, M. E.: Simulating the detailed chemical composition of secondary organic aerosol formed on a regional scale during the TORCH 2003 campaign in the southern UK, Atmos. Chem. Phys., 6, 419-431, doi: 10.5194/acp-6-419-2006, 2006a.

Johnson, D., Utembe, S. R., Jenkin, M. E., Derwent, R. G., Hayman, G. D., Alfarra, M. R., Coe, H., and McFiggans, G.: Simulating regional scale secondary organic aerosol formation during the TORCH 2003 campaign in the southern UK, Atmos. Chem. Phys., 6, 403-418, doi: 10.5194/acp-6-403-2006, 2006b.

Kanakidou, M., Seinfeld, J., Pandis, S., Barnes, I., Dentener, F., Facchini, M., Dingenen, R. V., Ervens, B., Nenes, A., and Nielsen, C.: Organic aerosol and global climate modelling: a review, Atmos. Chem. Phys., 5, 1053-1123, https://doi.org/10.5194/acp-5-1053-2005, 2005.

Kang, M., Fu, P., Kawamura, K., Yang, F., Zhang, H., Zang, Z., Ren, H., Ren, L., Zhao, Y., Sun, Y., and Wang, Z.: Characterization of biogenic primary and secondary organic aerosols in the marine atmosphere over the East China Sea, Atmos. Chem. Phys., 18, 13947-13967, doi: 10.5194/acp-18-13947-2018, 2018.

Kawamura, K., and Sakaguchi, F.: Molecular distributions of water soluble dicarboxylic acids in marine aerosols over the Pacific Ocean including tropics, J. Geophys. Res. Atmos., 104, 3501-3509, https://doi.org/10.1029/1998JD100041, 1999.

Kesselmeier, J., Bode, K., Gerlach, C., and Jork, E. M.: Exchange of atmospheric formic and acetic acids with trees and crop plants under controlled chamber and purified air conditions, Atmos. Environ., 32, 1765-1775, https://doi.org/10.1016/S1352-2310(97)00465-2, 1998.

Khare, P., Kumar, N., Kumari, K. M., and Srivastava, S. S.: Atmospheric formic and acetic acids: An overview, Rev. Geophys., 37, 227-248, doi: 10.1029/1998RG900005, 1999.

Kim, W., Lee, H., Kim, J., Jeong, U., and Kweon, J.: Estimation of seasonal diurnal variations

in primary and secondary organic carbon concentrations in the urban atmosphere: EC tracer and multiple regression approaches, Atmos. Environ., 56, 101-108, https://doi.org/10.1016/j.atmosenv.2012.03.076, 2012.

Kumar, A., Sudheer, A. K., Goswami, V., and Bhushan, R.: Influence of continental outflow on aerosol chemical characteristics over the Arabian Sea during winter, Atmos. Environ., 50, 182-191, https://doi.org/10.1016/j.atmosenv.2011.12.040, 2012.

Kumar, N., Kulshrestha, U. C., Saxena, A., Khare, P., Kumari, K. M., and Srivastava, S. S.: Formate and acetate levels compared in monsoon and winter rainwater at Dayalbagh, Agra (India), J. Atmos. Chem., 23, 81-87, doi: 10.1007/BF00058705, 1996.

Li, L., Ren, L., Ren, H., Yue, S., Xie, Q., Zhao, W., Kang, M., Li, J., Wang, Z., Sun, Y., and Fu, P.: Molecular Characterization and Seasonal Variation in Primary and Secondary Organic Aerosols in Beijing, China, J. Geophys. Res. Atmos., 123, 12,394-312,412, 10.1029/2018jd028527, 2018.

Liang, H., Lyu, L.-N., Sun, C., Ding, H., Wurgaft, E., and Yang, G.-P.: Low-molecular-weight organic acids as important factors impacting seawater acidification: A case study in the Jiaozhou Bay, China, Sci. Tot. Environ., 727, 138458, https://doi.org/10.1016/j.scitotenv.2020.138458, 2020.

Lim, H. J., and Turpin, B. J.: Origins of primary and secondary organic aerosol in Atlanta: results of time-resolved measurements during the Atlanta Supersite Experiment, Environ. Sci. Technol., 36, 4489-4496, 2002.

Lin, P., Hu, M., Deng, Z., Slanina, J., Han, S., Kondo, Y., Takegawa, N., Miyazaki, Y., Zhao, Y., and Sugimoto, N.: Seasonal and diurnal variations of organic carbon in PM2.5 in Beijing and the estimation of secondary organic carbon, J. Geophys. Res., 114, 10.1029/2008jd010902, 2009.

Liu, C.-N., Lin, S.-F., Awasthi, A., Tsai, C.-J., Wu, Y.-C., and Chen, C.-F.: Sampling and conditioning artifacts of PM2.5 in filter-based samplers, Atmos. Environ., 85, 48-53, https://doi.org/10.1016/j.atmosenv.2013.11.075, 2014a.

Liu, J., Zhang, X., Parker, E. T., Veres, P. R., Roberts, J. M., de Gouw, J. A., Hayes, P. L., Jimenez, J. L., Murphy, J. G., Ellis, R. A., Huey, L. G., and Weber, R. J.: On the gas-particle partitioning of soluble organic aerosol in two urban atmospheres with contrasting emissions: 2. Gas and particle phase formic acid, J. Geophys. Res. Atmos., 117, 10.1029/2012JD017912, 2012.

Liu, K., Duan, F., He, K., Ma, Y., and Cheng, Y.: Investigation on sampling artifacts of particle

associated PAHs using ozone denuder systems, Frontiers of Environmental Science & Engineering, 8, 284-292, doi: 10.1007/s11783-013-0555-7, 2014b.

Liu, Y., Sklorz, M., Schnelle-Kreis, J., Orasche, J., Ferge, T., Kettrup, A., and Zimmermann, R.: Oxidant denuder sampling for analysis of polycyclic aromatic hydrocarbons and their oxygenated derivates in ambient aerosol: Evaluation of sampling artefact, Chemosphere, 62, 1889-1898, https://doi.org/10.1016/j.chemosphere.2005.07.049, 2006.

Lonati, G., Ozgen, S., and Giugliano, M.: Primary and secondary carbonaceous species in PM2.5 samples in Milan (Italy), Atmos. Environ., 41, 4599-4610, https://doi.org/10.1016/j.atmosenv.2007.03.046, 2007.

Lyu, R., Shi, Z., Alam, M. S., Wu, X., Liu, D., Vu, T. V., Stark, C., Fu, P., Feng, Y., and Harrison, R. M.: Insight into the composition of organic compounds ($\geq$ C6) in PM2.5 in wintertime in Beijing, China, Atmos. Chem. Phys., 2019, 1-41, doi: 10.5194/acp-2018-1273, 2019.

Mikhailov, E. F., Mironova, S., Mironov, G., Vlasenko, S., Panov, A., Chi, X., Walter, D., Carbone, S., Artaxo, P., Heimann, M., Lavric, J., Pöschl, U., and Andreae, M. O.: Long-term measurements (2010–2014) of carbonaceous aerosol and carbon monoxide at the Zotino Tall Tower Observatory (ZOTTO) in central Siberia, Atmos. Chem. Phys., 17, 14365-14392, doi: 10.5194/acp-17-14365-2017, 2017.

Miyazaki, Y., Kondo, Y., Takegawa, N., Komazaki, Y., Fukuda, M., Kawamura, K., Mochida, M., Okuzawa, K., and Weber, R. J.: Time-resolved measurements of water-soluble organic carbon in Tokyo, J. Geophys. Res. Atmos., 111, 5573-5588, 2006.

Na, K., Sawant, A. A., Song, C., and Cocker, D. R.: Primary and secondary carbonaceous species in the atmosphere of Western Riverside County, California, Atmos. Environ., 38, 1345-1355, https://doi.org/10.1016/j.atmosenv.2003.11.023, 2004.

Pandis, S. N., Harley, R. A., Cass, G. R., and Seinfeld, J. H.: Secondary organic aerosol formation and transport, Atmospheric Environment. Part A. General Topics, 26, 2269-2282, https://doi.org/10.1016/0960-1686(92)90358-R, 1992.

Paulot, F., Wunch, D., Crounse, J. D., Toon, G. C., Millet, D. B., DeCarlo, P. F., Vigouroux, C., Deutscher, N. M., González Abad, G., Notholt, J., Warneke, T., Hannigan, J. W., Warneke, C., de Gouw, J. A., Dunlea, E. J., De Mazière, M., Griffith, D. W. T., Bernath, P., Jimenez, J. L., and Wennberg, P. O.: Importance of secondary sources in the atmospheric budgets of formic and acetic acids, Atmos. Chem. Phys., 11, 1989-2013, doi: 10.5194/acp-11-1989-2011, 2011.

Pio, C., Cerqueira, M., Harrison, R. M., Nunes, T., Mirante, F., Alves, C., Oliveira, C., Sanchez de la Campa, A., Artíñano, B., and Matos, M.: OC/EC ratio observations in Europe: Re-thinking

the approach for apportionment between primary and secondary organic carbon, Atmos. Environ., 45, 6121-6132, https://doi.org/10.1016/j.atmosenv.2011.08.045, 2011.

Ray, J., and McDow, S. R.: Dicarboxylic acid concentration trends and sampling artifacts, Atmos. Environ., 39, 7906-7919, https://doi.org/10.1016/j.atmosenv.2005.09.024, 2005.

Samara, C., Voutsa, D., Kouras, A., Eleftheriadis, K., Maggos, T., Saraga, D., and Petrakakis, M.: Organic and elemental carbon associated to PM10 and PM2.5 at urban sites of northern Greece, Environ. Sci. Pollut. R., 21, 1769-1785, doi: 10.1007/s11356-013-2052-8, 2014.

Sanderson, E. G., and Farant, J. P.: Atmospheric Size Distribution of PAHs: Evidence of a High-Volume Sampling Artifact, Environ. Sci. Technol., 39, 7631-7637, doi: 10.1021/es0510111, 2005.

Saturno, J., Holanda, B. A., Pöhlker, C., Ditas, F., Wang, Q., Moran-Zuloaga, D., Brito, J., Carbone, S., Cheng, Y., Chi, X., Ditas, J., Hoffmann, T., Hrabe de Angelis, I., Könemann, T., Lavrič, J. V., Ma, N., Ming, J., Paulsen, H., Pöhlker, M. L., Rizzo, L. V., Schlag, P., Su, H., Walter, D., Wolff, S., Zhang, Y., Artaxo, P., Pöschl, U., and Andreae, M. O.: Black and brown carbon over central Amazonia: long-term aerosol measurements at the ATTO site, Atmos. Chem. Phys., 18, 12817-12843, doi: 10.5194/acp-18-12817-2018, 2018.

Saxena, P., and Hildemann, L. M.: Water-soluble organics in atmospheric particles: a critical review of the literature and application of thermodynamics to identify candidate compounds, J. Atmos. Chem., 24, 57-109, doi: 10.1007/BF00053823, 1996.

Saylor, R. D., Edgerton, E. S., and Hartsell, B. E.: Linear regression techniques for use in the EC tracer method of secondary organic aerosol estimation, Atmos. Environ., 40, 7546-7556, https://doi.org/10.1016/j.atmosenv.2006.07.018, 2006.

Schauer, C., Niessner, R., and Pöschl, U.: Polycyclic aromatic hydrocarbons in urban air particulate matter: decadal and seasonal trends, chemical degradation, and sampling artifacts, Environ. Sci. Technol., 37, 2861-2868, doi: 10.1021/es034059s, 2003.

Schauer, J. J., Rogge, W. F., Hildemann, L. M., Mazurek, M. A., Cass, G. R., and Simoneit, B. R. T.: Source apportionment of airborne particulate matter using organic compounds as tracers, Atmos. Environ., 41, 241-259, https://doi.org/10.1016/j.atmosenv.2007.10.069, 2007.

Seguel A, R., Morales S, R. G. E., and Leiva G, M. A.: Estimations of primary and secondary organic carbon formation in PM2.5 aerosols of Santiago City, Chile, Atmos. Environ., 43, 2125-2131, https://doi.org/10.1016/j.atmosenv.2009.01.029, 2009.

Seinfeld, J. H., and Pandis, S. N.: Atmospheric Chemistry and Physics: From Air Pollution to Climate Change, 2rd edition, John Wiley & Sons, New York, 2006.

Shi, Z., Vu, T., Kotthaus, S., Grimmond, S., Harrison, R. M., Yue, S., Zhu, T., Lee, J., Han, Y., Demuzere, M., Dunmore, R. E., Ren, L., Liu, D., Wang, Y., Wild, O., Allan, J., Barlow, J., Beddows, D., Bloss, W. J., Carruthers, D., Carslaw, D. C., Chatzidiakou, L., Crilley, L., Coe, H., Dai, T., Doherty, R., Duan, F., Fu, P., Ge, B., Ge, M., Guan, D., Hamilton, J. F., He, K., Heal, M., Heard, D., Hewitt, C. N., Hu, M., Ji, D., Jiang, X., Jones, R., Kalberer, M., Kelly, F. J., Kramer, L., Langford, B., Lin, C., Lewis, A. C., Li, J., Li, W., Liu, H., Loh, M., Lu, K., Mann, G., McFiggans, G., Miller, M., Mills, G., Monk, P., Nemitz, E., O'Connor, F., Ouyang, B., Palmer, P. I., Percival, C., Popoola, O., Reeves, C., Rickard, A. R., Shao, L., Shi, G., Spracklen, D., Stevenson, D., Sun, Y., Sun, Z., Tao, S., Tong, S., Wang, Q., Wang, W., Wang, X., Wang, Z., Whalley, L., Wu, X., Wu, Z., Xie, P., Yang, F., Zhang, Q., Zhang, Y., Zhang, Y., and Zheng, M.: Introduction to Special Issue – In-depth study of air pollution sources and processes within Beijing and its surrounding region (APHH-Beijing), Atmos. Chem. Phys. Discuss., 2018, 1-62, 10.5194/acp-2018-922, 2018.

Sihabut, T., Ray, J., Northcross, A., and McDow, S. R.: Sampling artifact estimates for alkanes, hopanes, and aliphatic carboxylic acids, Atmos. Environ., 39, 6945-6956, https://doi.org/10.1016/j.atmosenv.2005.02.053, 2005.

Stavrakou, T., Müller, J. F., Peeters, J., Razavi, A., Clarisse, L., Clerbaux, C., Coheur, P. F., Hurtmans, D., De Mazière, M., Vigouroux, C., Deutscher, N. M., Griffith, D. W. T., Jones, N., and Paton-Walsh, C.: Satellite evidence for a large source of formic acid from boreal and tropical forests, Nat. Geosci., 5, 26-30, doi: 10.1038/ngeo1354, 2012.

Sun, Y., Zhang, Q., Zheng, M., Ding, X., Edgerton, E. S., and Wang, X.: Characterization and source apportionment of water-soluble organic matter in atmospheric fine particles (PM2.5) with high-resolution aerosol mass spectrometry and GC–MS, Environ. Sci. Technol., 45, 4854-4861, doi: 10.1021/es200162h, 2011.

Takegawa, N., Miyakawa, T., Kondo, Y., Jimenez, J. L., Zhang, Q., Worsnop, D. R., and Fukuda, M.: Seasonal and diurnal variations of submicron organic aerosol in Tokyo observed using the Aerodyne aerosol mass spectrometer, J. Geophys. Res. Atmos., 111, doi: 10.1029/2005JD006515, 2006.

Talbot, R. W., Beecher, K. M., Harriss, R. C., and Cofer Iii, W. R.: Atmospheric geochemistry of formic and acetic acids at a mid-latitude temperate site, J. Geophys. Res. Atmos., 93, 1638-1652, doi: 10.1029/JD093iD02p01638, 1988.

ten Brink, H., Maenhaut, W., Hitzenberger, R., Gnauk, T., Spindler, G., Even, A., Chi, X., Bauer, H., Puxbaum, H., Putaud, J.-P., Tursic, J., and Berner, A.: INTERCOMP2000: the comparability

of methods in use in Europe for measuring the carbon content of aerosol, Atmos. Environ., 38, 6507-6519, https://doi.org/10.1016/j.atmosenv.2004.08.027, 2004.

Trentmann, J., Yokelson, R. J., Hobbs, P. V., Winterrath, T., Christian, T. J., Andreae, M. O., and Mason, S. A.: An analysis of the chemical processes in the smoke plume from a savanna fire, J. Geophys. Res. Atmos., 110, doi: 10.1029/2004JD005628, 2005.

Tsapakis, M., and Stephanou, E. G.: Collection of gas and particle semi-volatile organic compounds: use of an oxidant denuder to minimize polycyclic aromatic hydrocarbons degradation during high-volume air sampling, Atmos. Environ., 37, 4935-4944, https://doi.org/10.1016/j.atmosenv.2003.08.026, 2003.

Turpin, B. J., and Huntzicker, J. J.: Identification of secondary organic aerosol episodes and quantitation of primary and secondary organic aerosol concentrations during SCAQS, Atmos. Environ., 29, 3527-3544, https://doi.org/10.1016/1352-2310(94)00276-Q, 1995.

Vecchi, R., Valli, G., Fermo, P., D'Alessandro, A., Piazzalunga, A., and Bernardoni, V.: Organic and inorganic sampling artefacts assessment, Atmos. Environ., 43, 1713-1720, https://doi.org/10.1016/j.atmosenv.2008.12.016, 2009.

Wang, G., Kawamura, K., Cheng, C., Li, J., Cao, J., Zhang, R., Zhang, T., Liu, S., and Zhao, Z.: Molecular distribution and stable carbon isotopic composition of dicarboxylic acids, ketocarboxylic acids, and α-dicarbonyls in size-resolved atmospheric particles from Xi'an City, China, Environ. Sci. Technol., 46, 4783-4791, doi: 10.1021/es204322c, 2012.

Wang, G., Zhang, R., Gomez, M. E., Yang, L., Levy Zamora, M., Hu, M., Lin, Y., Peng, J., Guo, S., Meng, J., Li, J., Cheng, C., Hu, T., Ren, Y., Wang, Y., Gao, J., Cao, J., An, Z., Zhou, W., Li, G., Wang, J., Tian, P., Marrero-Ortiz, W., Secrest, J., Du, Z., Zheng, J., Shang, D., Zeng, L., Shao, M., Wang, W., Huang, Y., Wang, Y., Zhu, Y., Li, Y., Hu, J., Pan, B., Cai, L., Cheng, Y., Ji, Y., Zhang, F., Rosenfeld, D., Liss, P. S., Duce, R. A., Kolb, C. E., and Molina, M. J.: Persistent sulfate formation from London Fog to Chinese haze, Proc. Nat. Acad. Sci. U.S.A., 113, 13630-13635, 10.1073/pnas.1616540113, 2016.

Wang, H., Kawamura, K., and Yamazaki, K.: Water-soluble dicarboxylic acids, ketoacids and dicarbonyls in the atmospheric aerosols over the Southern Ocean and western Pacific Ocean, J. Atmos. Chem., 53, 43-61, doi: 10.1007/s10874-006-1479-4, 2006.

Wang, Q., He, X., Huang, X. H. H., Griffith, S. M., Feng, Y., Zhang, T., Zhang, Q., Wu, D., and Yu, J. Z.: Impact of Secondary Organic Aerosol Tracers on Tracer-Based Source Apportionment of Organic Carbon and PM2.5: A Case Study in the Pearl River Delta, China, ACS Earth Space Chem., 1, 562-571, doi: 10.1021/acsearthspacechem.7b00088, 2017.

Wang, Y., Zhuang, G., Chen, S., An, Z., and Zheng, A.: Characteristics and sources of formic, acetic and oxalic acids in PM2.5 and PM10 aerosols in Beijing, China, Atmos. Res., 84, 169-181, https://doi.org/10.1016/j.atmosres.2006.07.001, 2007.

Watson, J. G., Chow, J. C., Chen, L. W. A., and Frank, N. H.: Methods to assess carbonaceous aerosol sampling artifacts for IMPROVE and other long-term Networks, Journal of the Air & Waste Management Association, 59, 898-911, doi: 10.3155/1047-3289.59.8.898, 2009.

Weber, R. J., Sullivan, A. P., Peltier, R. E., Russell, A., Yan, B., Zheng, M., de Gouw, J., Warneke, C., Brock, C., Holloway, J. S., Atlas, E. L., and Edgerton, E.: A study of secondary organic aerosol formation in the anthropogenic-influenced southeastern United States, J. Geophys. Res. Atmos., 112, 10.1029/2007JD008408, 2007.

Willey, J. D., and Wilson, C. A.: Formic and acetic acids in atmospheric condensate in Wilmington, North Carolina, J. Atmos. Chem., 16, 123-133, doi: 10.1007/BF00702782, 1993.

Wu, C., and Yu, J. Z.: Determination of primary combustion source organic carbon-to-elemental carbon (OC/EC) ratio using ambient OC and EC measurements: secondary OC-EC correlation minimization method, Atmos. Chem. Phys., 16, 5453-5465, doi: 10.5194/acp-16-5453-2016, 2016.

Wu, L., Ren, H., Wang, P., Chen, J., Fang, Y., Hu, W., Ren, L., Deng, J., Song, Y., Li, J., Sun, Y., Wang, Z., Liu, C.-Q., Ying, Q., and Fu, P.: Aerosol ammonium in the urban boundary layer in Beijing: Insights from nitrogen isotope ratios and simulations in summer 2015, Environ. Sci. Technol. Lett., 6, 389-395, doi: 10.1021/acs.estlett.9b00328, 2019.

Xu, G., Lee, X., and Lv, Y.: Urban and rural observations of carboxylic acids in rainwater in Southwest of China: the impact of urbanization, J. Atmos. Chem., 62, 249-260, doi: 10.1007/s10874-010-9151-4, 2009.

Yu, X. Y., Cary, R. A., and Laulainen, N. S.: Primary and secondary organic carbon downwind of Mexico City, Atmos. Chem. Phys., 9, 6793-6814, doi: 10.5194/acp-9-6793-2009, 2009.

Yuan, Z. B., Yu, J. Z., Lau, A. K. H., Louie, P. K. K., and Fung, J. C. H.: Application of positive matrix factorization in estimating aerosol secondary organic carbon in Hong Kong and its relationship with secondary sulfate, Atmos. Chem. Phys., 6, 25-34, doi: 10.5194/acp-6-25-2006, 2006.

Yue, S., Ren, L., Song, T., Li, L., Xie, Q., Li, W., Kang, M., Zhao, W., Wei, L., Ren, H., Sun, Y., Wang, Z., Ellam, R. M., Liu, C.-Q., Kawamura, K., and Fu, P.: Abundance and diurnal trends of fluorescent bioaerosols in the troposphere over Mt. Tai, China, in spring, J. Geophys. Res. Atmos., 124, 4158-4173, doi: 10.1029/2018jd029486, 2019.

Zhang, Q., Worsnop, D. R., Canagaratna, M. R., and Jimenez, J. L.: Hydrocarbon-like and oxygenated organic aerosols in Pittsburgh: insights into sources and processes of organic aerosols, Atmos. Chem. Phys., 5, 3289-3311, doi: 10.5194/acp-5-3289-2005, 2005.

Zhang, R., Wang, G., Song, G., Zamora, M. L., Qi, Y., Yun, L., Wang, W., Min, H., and Yuan, W.: Formation of urban fine particulate matter, Chem. Rev., 115, 3803-3855, doi: 10.1021/acs.chemrev.5b00067, 2015.

Zhang, Y., Ren, H., Sun, Y., Cao, F., Chang, Y., Liu, S., Lee, X., Agrios, K., Kawamura, K., Liu, D., Ren, L., Du, W., Wang, Z., Prévôt, A. S. H., Szidat, S., and Fu, P.: High contribution of nonfossil sources to submicrometer organic aerosols in Beijing, China, Environ. Sci. Technol., 51, 7842-7852, doi: 10.1021/acs.est.7b01517, 2017.

Zhang, Y.-L., Li, J., Zhang, G., Zotter, P., Huang, R.-J., Tang, J.-H., Wacker, L., Prévôt, A. S. H., and Szidat, S.: Radiocarbon-Based Source Apportionment of Carbonaceous Aerosols at a Regional Background Site on Hainan Island, South China, Environ. Sci. Technol., 48, 2651-2659, doi: 10.1021/es4050852, 2014.

Zhang, Y. L., Lee, X. Q., and Cao, F.: Chemical characteristics and sources of organic acids in precipitation at a semi-urban site in Southwest China, Atmos. Environ., 45, 413-419, https://doi.org/10.1016/j.atmosenv.2010.09.067, 2011.

Zhang, Y. L., Kawamura, K., Fu, P. Q., Boreddy, S. K. R., Watanabe, T., Hatakeyama, S., Takami, A., and Wang, W.: Aircraft observations of water-soluble dicarboxylic acids in the aerosols over China, Atmos. Chem. Phys., 16, 6407-6419, doi: 10.5194/acp-16-6407-2016, 2016.

Zhao, X. J., Zhao, P. S., Xu, J., Meng, W., Pu, W. W., Dong, F., He, D., and Shi, Q. F.: Analysis of a winter regional haze event and its formation mechanism in the North China Plain, Atmos. Chem. Phys., 13, 5685-5696, doi: 10.5194/acp-13-5685-2013, 2013.

Zheng, M., Cass, G. R., Schauer, J. J., and Edgerton, E. S.: Source Apportionment of PM2.5 in the Southeastern United States Using Solvent-Extractable Organic Compounds as Tracers, Environ. Sci. Technol., 36, 2361-2371, doi: 10.1021/es011275x, 2002.

Zhou, S., Wang, Z., Gao, R., Xue, L., Yuan, C., Wang, T., Gao, X., Wang, X., Nie, W., Xu, Z., Zhang, Q., and Wang, W.: Formation of secondary organic carbon and long-range transport of carbonaceous aerosols at Mount Heng in South China, Atmos. Environ., 63, 203-212, https://doi.org/10.1016/j.atmosenv.2012.09.021, 2012.

Zhou, W., Sun, Y., Xu, W., Zhao, X., Wang, Q., Tang, G., Zhou, L., Chen, C., Du, W., Zhao, J., Xie, C., Fu, P., and Wang, Z.: Vertical Characterization of Aerosol Particle Composition in

Beijing, China: Insights From 3-Month Measurements With Two Aerosol Mass Spectrometers, J. Geophys. Res. Atmos., 123, 13,016-013,029, doi: 10.1029/2018JD029337, 2018a.

Zhou, W., Zhao, J., Ouyang, B., Mehra, A., Xu, W., Wang, Y., Bannan, T. J., Worrall, S. D., Priestley, M., Bacak, A., Chen, Q., Xie, C., Wang, Q., Wang, J., Du, W., Zhang, Y., Ge, X., Ye, P., Lee, J. D., Fu, P., Wang, Z., Worsnop, D., Jones, R., Percival, C. J., Coe, H., and Sun, Y.: Production of N2O5 and ClNO2 in summer in urban Beijing, China, Atmos. Chem. Phys., 18, 11581-11597, 10.5194/acp-18-11581-2018, 2018b.

Zhu, C.-S., Tsai, C.-J., Chen, S.-C., Cao, J.-J., and Roam, G.-D.: Positive sampling artifacts of organic carbon fractions for fine particles and nanoparticles in a tunnel environment, Atmos. Environ., 54, 225-230, https://doi.org/10.1016/j.atmosenv.2012.02.060, 2012.

Zotter, P., El-Haddad, I., Zhang, Y., Hayes, P. L., Zhang, X., Lin, Y.-H., Wacker, L., Schnelle-Kreis, J., Abbaszade, G., Zimmermann, R., Surratt, J. D., Weber, R., Jimenez, J. L., Szidat, S., Baltensperger, U., and Prévôt, A. S. H.: Diurnal cycle of fossil and nonfossil carbon using radiocarbon analyses during CalNex, J. Geophys. Res. Atmos., 119, 6818-6835, doi: 10.1002/2013JD021114, 2014.

---

## Author Comment (AC2) · 18 Jun 2020

**Responses to Reviewer #2**

We appreciate this reviewer for the useful comments and suggestions, which greatly improve the quality of the manuscript. We revised the MS accordingly. The point-to-point responses to all the comments are given below in blue.

The manuscript studied the vertical distribution of particle-phase dicarboxylic acids, oxoacids, and α-dicarbonyls in urban Beijing during the 2015 Victory Parade period based on the 325-meter tower by using GC/MS, Ion Chromatography, and OC/EC analyzer. This study showed that concentrations of oxalic acid at 120 m and 260 m were more abundant than that at 8 m during the sampling period because of higher oxidation at high altitude. Vehicular exhausts were demonstrated as the main contributor for phthalic acid. Although this study did many correlation analyses in section 3, simple correlation analyses didn't effectively evidence the sources and formation of species such as oxalic acid. In addition, many results and conclusions in section 3 were concluded with indefinite words such as may, causing that the conclusions were too speculative. Therefore, the authors should provide some stronger evidences in this section. This study further indicated the sources of organic acids based on meteorological parameters and FLEXPART-WRF model analyses. However, the source analysis at each sampling altitude was not showed. Is there any difference in the source of organic acids at each altitude? Because the study conducted on the vertical measurement, the data is valuable compared to the ground base. I might ask the author to make significant improvement on this paper before they can be accepted in the ACP.

**Response:** The real atmospheric environment is complex, where diverse formation mechanisms of organic aerosols exist. Field campaign estimates the dominant mechanism of aerosol components based on the atmospheric variables, which can not reveal the detailed mechanisms of organic compounds like smog chamber study. Owing to the insufficient data, the PMF analysis run at each altitude existed high uncertainties, thus the PMF source apportionment analysis is to the overall results in our study.

The sample numbers will be considered to increase at each altitude. Simultaneously, more

analyses will be employed to the aerosol samples to better estimate the relative contribution of sources and secondary process at different height.

This paper firstly investigates the vertical distribution of diacids, oxoacids and α-dicarbonyls in Beijing, analyzes the primary sources and secondary processes in clean and polluted days, and demonstrates the feedback of organic acids under the control of anthropogenic emissions. Such measurements in the troposphere are also critical for estimating the regional transport to air quality in Beijing, which are favorable for chemical transport models to better evaluate the vertical contributions of local emissions, regional transport, aqueous and photochemical oxidation processes. Furthermore, the synchronous measurements, like Chemical Ionization Mass Spectrometer, is considered to use in combination with the sampling of diacids, oxoacids and α-dicarbonyls in gas and particle phase to better understand the gas-particle transformation of these acids at different heights.

P13 L21: How to exclude the contribution of accumulation of local emissions? Other evidences of aqueous-phase oxidation also should be provided.

**Response:** The local emissions also contributed to diacids and related compounds in polluted days. Owing to the increase of relative humidity in the low troposphere, the aqueous formation of oxalic acid enhanced in the polluted days, which was more important than local primary emissions.

Hydrated glyoxal (Gly) and methyglyoxal (MeGly) can ultimately produce oxalic acid ($C_2$) via the formation of glyoxylic ($\omega C_2$) and pyruvic (Pyr) acids as intermediates (Carlton et al., 2007;Carlton et al., 2009;Tan et al., 2010). The concentration ratio of relative abundance of $C_2$ in total diacids ($C_2$/total diacids) is known as a useful marker to assess the aerosol oxidation level, because $C_2$ is the end product mostly formed via the oxidation of longer carbon-chain diacids and other precursors in the atmosphere (Kawamura and Bikkina, 2016). Therefore, the relationships for $C_2/\omega C_2$, $C_2$/Pyr, $C_2$/Gly and $C_2$/MeGly with $C_2$/total diacids were applied to better understand the aqueous formation of $C_2$. Compared to clean days, good coefficients were obtained for $C_2/\omega C_2$, $C_2$/Pyr, $C_2$/Gly and $C_2$/MeGly with $C_2$/total diacids in polluted days, which generally decreased with the sampling height. This phenomenon suggested that the vertical transformations of $\omega C_2$, Pyr, Gly and MeGly

to form $C_2$ were more clearly observed at the ground surface due to the higher relative humidity. It's worth noting that the value order of relative humidity was 8 m > 120 m > 260 m (Fig. S1).

Furthermore, the relative content of $C_2$ in water-soluble organic carbon ($C_2$-C/WSOC) showed different vertical distributions and variations in clean and polluted days. Aged organic aerosols are usually characterized by the larger contribution of oxalic acid to WSOC ($C_2$-C/WSOC). For example, $C_2$-C/WSOC ratio was higher in the photochemically aged aerosols collected at Hong Kong (6.8%) (Ho et al., 2011) and Mount Hua (6.3%) (Meng et al., 2014) compared with the ratio (0.17%) in Ulaanbaatar aerosols that are significantly affected by substantial anthropogenic emissions (Jung et al., 2010). Due to the high temperature and relative humidity, the photochemical reaction is active in Hong Kong (Ho et al., 2011). Mount Hua is the highest mountain in central China and is a typically isolated site to investigate the atmospheric long-range transport of organic compounds (Meng et al., 2014). In contrast, diacids and related compounds in winter were mainly associated with uncontrolled wastes plastic burning, coal power plants and vehicular emissions in Ulaanbaatar (Jung et al., 2010). Generally, in clean days, the C2-C/WSOC ratio showed relatively large values at upper heights in this study (Fig. 9a). Moreover, in the transition from clean to polluted days, the C2-C/WSOC ratio values at the ground level, 120 m and 260 m slightly increased. However, in the more polluted days, $C_2$-C/WSOC ratios at the ground level were obviously higher than those at 120 m and 260 m owing to the accumulation of pollutants and moisture in ground surface atmosphere, which also supported the conclusion that the increased aqueous-phase oxidation may be a major source of oxalic acid in polluted days.

P4 L21: why were the blank samples only collected for half a minute? I saw that $PM_{2.5}$ samples were collected for 23 h.

**Response:** The field blank is sampled to see whether the aerosol samples have been polluted during the operation process, including the placing and collecting processes of the filter, which takes a few seconds. This sampling procedure doesn't aim to see the environmental impact during sampling time. Please see the definition of field blank at

website (https://www.lcslaboratory.com/field-blank/).

P5 L6: Grammatically something wrong.

**Response:** Corrected.

P8 L4: How to exclude other sources (e.g., long-range transport, aqueous or heterogeneous reactions) for SOC?

**Response:** We have modified the sentence to avoid contradiction. "The SOC/POC ratios at 120 m (1.8 ± 0.79) and 260 m (1.9 ± 0.92) were higher than those at the ground level (0.51 ± 0.3) (Table. S1), demonstrating that more aged aerosols accumulated at upper layers (Fig. 2f)."

P8 L10: other sources should be discussed. Please see the above comment.

**Response:** We have modified the sentence in the revised manuscript. "These vertical phenomena were also observed for total oxoacids and α-dicarbonyls, suggesting that the aging level of diacids and related compounds slightly increased at 260 m of the atmosphere."

P12 L16: $C_3/C_4$ should be $C_2/C_3$?

**Response:** Corrected.

P13 L9: What is the definition criteria of polluted episodes?

**Response:** The pollution level is defined by air quality index (AQI), and is classified as light (101 − 150), moderate (151 − 200), heavy (201 − 300) and extremely heavy (301 − 500) pollution by Chinese Environmental Protection Ministry. Zhao et al. (2017) has also investigated chemical composition and diurnal variations of submicron aerosol species at ground level and 260 m during same polluted episodes in overlapped sampling time.

P14 L15-16: Definition of pollution level should be provided.

**Response:** The pollution level is defined by air quality index (AQI) according to local report from environmental monitor station. AQI is calculated from the following equation:

$$\text{AQI} = \frac{AQI_h - AQI_l}{C_h - C_l}(C - C_l) + AQI_l$$

C is the pollutant concentration. $AQI_l$ and $AQI_h$ are AQI values corresponding to $C_l$ and $C_h$, respectively. $C_l$ and $C_h$ are lower and upper limits near the pollutant concentration. $AQI_l$, $AQI_h$, $C_l$ and $C_h$ are constants, which can be obtained from the document of Technical Regulation on Ambient Air Quality Index (HJ 633 − 2012) issued by Chinese Environmental Protection Ministry. AQI is classified as light (101 − 150), moderate (151 − 200), heavy (201 − 300) and extremely heavy (301 − 500) pollution by Chinese Environmental Protection Ministry. We have added the definition of pollution level in the manuscript. Please see lines 9 − 10 in page 13.

P17 L8: Was the coal combustion included in the anthropogenic emissions? Maybe need related literature?

**Response:** Zhang et al. (2008) found that the saturated *n*-diacids ($C_3 − C_{10}$), unsaturated diacids (fumaric, maleic, phthalic, isophthalic and terephthalic acids) and related organic precursors, such as n-alkanes, PAHs and unsaturated fatty acids, can be directly emitted by industrial and residential coal combustion in China. Guo et al. (2013) reported that the contribution of coal combustion was associated with cooking in urban outskirts and rural areas during Beijing 2008 Olympics. Coal combustion accounted for $5.8 \pm 5.5\%$ and $7.8 \pm 4.6\%$ of the measured OC under the control on anthropogenic emissions (Guo et al., 2013), which was similar to the result in our study. The uncontrolled coal combustion in the rural areas was an important source to hopanes and PAHs in Beijing (Yu et al., 2018; Ren et al., 2018; Guo et al., 2013).

Coal combustion is an important anthropogenic source (Zhu et al., 2018), which largely influence the mass concentrations of organic compounds in China, especially in winter (Wang et al., 2012; Wang et al., 2006; Yu et al., 2018; Ren et al., 2018; Sun et al., 2016; Sun et al., 2013; Wang et al., 2019). We have added the reference and corrected the mistake in the revised manuscript. Please see lines 7 − 9 in page 17.

P17 L22: Same question as above.

**Response:** We have added the reference and corrected the mistake in the revised manuscript. "In this paper, PMF analysis showed that the contributed fraction of anthropogenic emissions (49 − 55%), including biomass burning, motor vehicles and coal combustion (Zhu et al., 2018), to diacids and related compounds were larger than that of secondary formation pathways (37 − 44%)." Please see lines 21 − 23 in page 17.

**References**

Carlton, A. G., Turpin, B. J., Altieri, K. E., Seitzinger, S., Reff, A., Lim, H.-J., and Ervens, B.: Atmospheric oxalic acid and SOA production from glyoxal: Results of aqueous photooxidation experiments, Atmos. Environ., 41, 7588-7602, https://doi.org/10.1016/j.atmosenv.2007.05.035, 2007.

Carlton, A. G., Wiedinmyer, C., and Kroll, J. H.: A review of secondary organic aerosol (SOA) formation from isoprene, Atmos. Chem. Phys., 9, 4987-5005, https://doi.org/10.5194/acp-9-4987-2009, 2009.

Guo, H., Ling, Z. H., Cheung, K., and Jiang, F.: Characterization of photochemical pollution at different elevations in mountainous areas in Hong Kong, Atmos. Chem. Phys., 13, 3881-3898, https://doi.org/10.5194/acp-13-3881-2013, 2013.

Ho, K. F., Ho, S. S. H., Lee, S. C., Kawamura, K., Zou, S. C., Cao, J. J., and Xu, H. M.: Summer and winter variations of dicarboxylic acids, fatty acids and benzoic acid in PM2.5 in Pearl Delta River Region, China, Atmos. Chem. Phys., 10, 26677-26703, 2011.

Jung, J., Tsatsral, B., Kim, Y. J., and Kawamura, K.: Organic and inorganic aerosol compositions in Ulaanbaatar, Mongolia, during the cold winter of 2007 to 2008: Dicarboxylic acids, ketocarboxylic acids, andα-dicarbonyls, J. Geophys. Res., 115, 10.1029/2010jd014339, 2010.

Meng, J., Wang, G., Li, J., Cheng, C., Ren, Y., Huang, Y., Cheng, Y., Cao, J., and Zhang, T.: Seasonal characteristics of oxalic acid and related SOA in the free troposphere of Mt. Hua, central China: implications for sources and formation mechanisms, Sci. Tot. Environ., 493, 1088-1097, https://doi.org/10.1016/j.scitotenv.2014.04.086, 2014.

Ren, H., Kang, M., Ren, L., Zhao, Y., Pan, X., Yue, S., Li, L., Zhao, W., Wei, L., Xie, Q.,

Li, J., Wang, Z., Sun, Y., Kawamura, K., and Fu, P.: The organic molecular composition, diurnal variation, and stable carbon isotope ratios of PM2.5 in Beijing during the 2014 APEC summit, Environ. Pollut., 243, 919-928, https://doi.org/10.1016/j.envpol.2018.08.094, 2018.

Sun, Y., Wang, Z., Fu, P., Yang, T., Jiang, Q., Dong, H., Li, J., and Jia, J.: Aerosol composition, sources and processes during wintertime in Beijing, China, Atmos. Chem. Phys., 13, 4577-4592, https://doi.org/10.5194/acp-13-4577-2013, 2013.

Sun, Y., Wang, Z., Wild, O., Xu, W., Chen, C., Fu, P., Du, W., Zhou, L., Zhang, Q., Han, T., Wang, Q., Pan, X., Zheng, H., Li, J., Guo, X., Liu, J., and Worsnop, D. R.: "APEC Blue": secondary aerosol reductions from emission controls in Beijing, Sci. Rep., 6, 20668, doi: 10.1038/srep20668, 2016.

Tan, Y., Carlton, A. G., Seitzinger, S. P., and Turpin, B. J.: SOA from methylglyoxal in clouds and wet aerosols: Measurement and prediction of key products, Atmos. Environ., 44, 5218-5226, https://doi.org/10.1016/j.atmosenv.2010.08.045, 2010.

Wang, G., Kawamura, K., Lee, S., Ho, K., and Cao, J.: Molecular, Seasonal, and Spatial Distributions of Organic Aerosols from Fourteen Chinese Cities, Environ. Sci. Technol., 40, 4619-4625, doi: 10.1021/es060291x, 2006.

Wang, G., Kawamura, K., Cheng, C., Li, J., Cao, J., Zhang, R., Zhang, T., Liu, S., and Zhao, Z.: Molecular distribution and stable carbon isotopic composition of dicarboxylic acids, ketocarboxylic acids, and α-dicarbonyls in size-resolved atmospheric particles from Xi'an City, China, Environ. Sci. Technol., 46, 4783-4791, doi: 10.1021/es204322c, 2012.

Wang, J., Liu, D., Ge, X., Wu, Y., Shen, F., Chen, M., Zhao, J., Xie, C., Wang, Q., Xu, W., Zhang, J., Hu, J., Allan, J., Joshi, R., Fu, P., Coe, H., and Sun, Y.: Characterization of black carbon-containing fine particles in Beijing during wintertime, Atmos. Chem. Phys., 19, 447-458, 10.5194/acp-19-447-2019, 2019.

Yu, Q., Yang, W., Zhu, M., Gao, B., Li, S., Li, G., Fang, H., Zhou, H., Zhang, H., Wu, Z., Song, W., Tan, J., Zhang, Y., Bi, X., Chen, L., and Wang, X.: Ambient PM2.5-bound polycyclic aromatic hydrocarbons (PAHs) in rural Beijing: Unabated with enhanced temporary emission control during the 2014 APEC summit and largely aggravated after the start of wintertime heating, Environ. Pollut., 238, 532-542,

https://doi.org/10.1016/j.envpol.2018.03.079, 2018.

Zhang, Y., Schauer, J. J., Zhang, Y., Zeng, L., Wei, Y., Liu, Y., and Shao, M.: Characteristics of Particulate Carbon Emissions from Real-World Chinese Coal Combustion, Environ. Sci. Technol., 42, 5068-5073, doi: 10.1021/es7022576, 2008.

Zhao, J., Du, W., Zhang, Y., Wang, Q., Chen, C., Xu, W., Han, T., Wang, Y., Fu, P., Wang, Z., Li, Z., and Sun, Y.: Insights into aerosol chemistry during the 2015 China Victory Day parade: results from simultaneous measurements at ground level and 260 m in Beijing, Atmos. Chem. Phys., 17, 3215-3232, doi: 10.5194/acp-17-3215-2017, 2017.

Zhu, Y., Huang, L., Li, J., Ying, Q., Zhang, H., Liu, X., Liao, H., Li, N., Liu, Z., Mao, Y., Fang, H., and Hu, J.: Sources of particulate matter in China: Insights from source apportionment studies published in 1987–2017, Environment International, 115, 343-357, https://doi.org/10.1016/j.envint.2018.03.037, 2018.

---

## Author Response (AR1)

[revised manuscript text omitted]

BDL: below detection limit, which is ca. 0.005 ng  $m^{-3}$  for the target compounds.

Table 3. Abundance and naming of measured ions ( $\mu g m^{-3}$ ) and organic tracers ( $ng m^{-3}$ ) used in the PMF analysis.

| Tracers                      | Grouping                                               | Sources                     | Mean/SD          |           |           |
|------------------------------|--------------------------------------------------------|-----------------------------|------------------|-----------|-----------|
|                              |                                                        |                             | The ground level | 120 m     | 260 m     |
| PAHs276                      | indeno[1,2,3-cd]pyrene,                                | Combustion sources          | 0.41/0.23        | 0.24/0.18 | 0.08/0.06 |
|                              | benzo[ghi]perylene                                     | (mainly coal combustion)    |                  |           |           |
| Levoglucosan                 |                                                        | Biomass burning             | 19/16            | 21/14     | 23/15     |
| Hopanes                      | $\alpha\beta$ -hopane, $\alpha\beta S\&R$ -homohopane, | Fossil fuel combustion      | 1.5/0.5          | 2.0/1.0   | 1.0/0.6   |
|                              | $\alpha\beta S\&R$ -bishomohopane                      | (e.g. vehicle exhaust,      |                  |           |           |
|                              |                                                        | coal combustion)            |                  |           |           |
| Isoprene SOA                 | 2-methylglyceric acid,                                 | Isoprene derived SOA,       | 31/20            | 41/38     | 48/37     |
| tracers                      | 2-methylthreitol,                                      | plants emissions            |                  |           |           |
|                              | 2-methylerythritol,                                    |                             |                  |           |           |
|                              | C 5 -alkene triols                          |                             |                  |           |           |
| $SO_4^{2-}$                  |                                                        | Secondary sulfate formation | 36/34            | 43/46     | 51/46     |
| NO 3 - |                                                        | Secondary nitrate formation | 20/19            | 29/30     | 36/43     |